# ENTROPY-BASED UNCERTAINTY MODELING FOR TRAJECTORY PREDICTION IN AUTONOMOUS DRIVING

## ABSTRACT

In autonomous driving, accurate motion prediction is essential for safe and efficient motion planning. To ensure safety, planners must rely on reliable uncertainty information about the predicted future behavior of surrounding agents, yet this aspect has received limited attention. This paper addresses the so-far neglected problem of uncertainty modeling in trajectory prediction. We adopt a holistic approach that focuses on uncertainty quantification, decomposition, and the influence of model composition. Our method is based on a theoretically grounded information-theoretic approach to measure uncertainty, allowing us to decompose total uncertainty into its aleatoric and epistemic components. We conduct extensive experiments on the nuScenes dataset to assess how different model architectures and configurations affect uncertainty quantification and model robustness. Our analysis thoroughly explores the uncertainty quantification capabilities of several state-of-the-art prediction models, examining the relationship between uncertainty and prediction error in both in- and out-of-distribution scenarios, as well as robustness in out-of-distribution.

## 1 INTRODUCTION

In a machine learning driven Autonomous Driving (AD) stack, motion prediction connects the upstream task of environment perception with the downstream task of ego-motion planning (Hu et al., 2023). The role of a motion predictor is to infer the future motion of traffic agents relevant to the ego, ensuring safe and efficient progress toward a goal (Hagedorn et al., 2024). To achieve this, a predictor must tackle several challenges such as imperfect perception, complex interactions between traffic agents, as well as the multitude of distinct potential actions each agent might undertake, motivated by different goals. Such challenges drive the need to consider the problem in a probabilistic manner and incorporate uncertainty into prediction outputs. Accurately quantifying prediction uncertainty is essential for ensuring interpretability and building trust in the overall system.

In the AD community, the future motion of surrounding traffic agents is often modeled in the form of trajectories. Thus, probabilistic trajectory prediction involves capturing a distribution $p(y|x, \mathcal{D})$ of future trajectories $y$ conditioned on contextual data $x$ and a dataset $\mathcal{D}$. Contextual data $x$ usually contains past trajectories of surrounding agents and map information. There are different strategies for capturing this highly multi-modal distribution owing to the distinct actions or goals. Some methods attempt to directly predict the modes of the distribution along with their associated weights (Gao et al., 2020; Kim et al., 2021; Deo et al., 2022). Others use a parametric mixture distribution, such as a Gaussian Mixture Model (GMM), where the modes correspond to the predicted trajectories (Tolstaya et al., 2021; Varadarajan et al., 2022; Liu et al., 2024; Look et al., 2023). Alternatively, generative trajectory prediction models use well-known autoencoder or diffusion architectures to model latent variables and draw trajectory samples (Salzmann et al., 2020; Seff et al., 2023; Janjoš et al., 2023a; Jiang et al., 2023).

The majority of approaches for modeling the distribution of future trajectories, $p(y|x, \mathcal{D})$, in AD rely on neural networks. They are often underspecified by the available data, meaning that no single parameter configuration is favored. When considering uncertainty in the model parameters, the predictive distribution (Kapoor et al., 2022; MacKay, 1992) over future trajectories is computed as

$$p(y|x, \mathcal{D}) = \int p(y|x, \mathcal{W})p(\mathcal{W}|\mathcal{D})d\mathcal{W} \approx \int p(y|x, \mathcal{W})q(\mathcal{W})d\mathcal{W} \,, \tag{1}$$

where $\mathcal{W}$ represents the neural network weights, and $p(\mathcal{W}|\mathcal{D})$ represents the posterior distribution. The predictive distribution represents a Bayesian model average, meaning that instead of relying on a single hypothesis with a specific set of parameters, it considers all possible parameter configurations, weighted by their posterior $p(\mathcal{W}|\mathcal{D})$. This marginalization process removes the reliance on a single weight configuration in the predictive distribution, resulting in better calibration and accuracy compared to traditional training methods (Wilson & Izmailov, 2020). Since the exact posterior is often intractable, various approximations $q(\mathcal{W})$ have been developed, such as variational inference (Graves, 2011), Dropout (Gal & Ghahramani, 2016), Laplace approximation (Ritter et al., 2018), deep ensembles (Lakshminarayanan et al., 2017), or Markov Chain Monte Carlo (MCMC) methods (Welling & Teh, 2011).

Despite many successful approaches of approximating the posterior distribution, the AD prediction community has not yet tried to quantify or to decompose the uncertainty of trajectory prediction models in a theoretically principled manner (Wilson, 2020). This gap is notable, especially considering the potential benefits of decomposing the uncertainty. The total uncertainty can be decomposed into two types: aleatoric and epistemic uncertainty (Wimmer et al., 2023; Hüllermeier, 2021). Aleatoric uncertainty represents the inherent variability within the data, such as the equal likelihood of a vehicle turning left or right at a T-junction. This type of uncertainty cannot be reduced, even with more data. In contrast, epistemic uncertainty arises from the lack of knowledge or information and can be reduced by collecting more data (Wimmer et al., 2023). Knowledge of epistemic uncertainty is helpful in various contexts, e.g. risk-sensitive reinforcement learning (Depeweg et al., 2018) and Out-of-Distribution (OOD) detection (Amini et al., 2020). By understanding the sources of uncertainty, one can confirm that an autonomous vehicle finds itself in an OOD scenario by observing a higher epistemic uncertainty. This can be an important signal for a planner that uses predictions in its decision-making. By incorporating this information, planners can make more informed decisions and potentially take preventative actions in situations of high uncertainty.

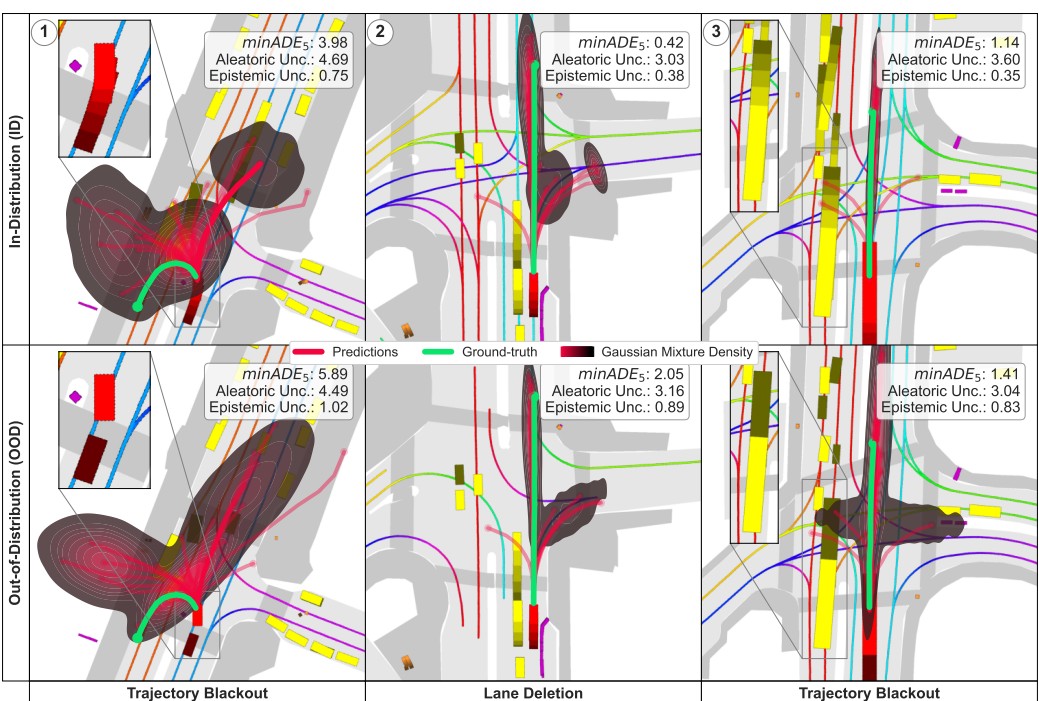

Figure 1: The predictive distribution $p(y|x, \mathcal{D})$ of future trajectories for three example scenarios. The first row shows in-distribution scenarios, while the second row presents OOD cases: in ① and ③, segments of the input history have been removed, while in ②, parts of lane information have been removed. Both alterations mimic perception malfunctions. Naturally, prediction error is higher in the second row, indicated by the higher Minimum Average Displacement Error (minADE) metric, see Sec. 3 for details. Generally, we observe a correlation between minADE and total uncertainty. In these examples, epistemic uncertainty serves as a useful indicator for detecting OOD scenarios.

In this paper, we address the challenge of modeling the uncertainty of trajectory prediction models within the autonomous driving (AD) domain from a holistic perspective. We focus on the quantification and decomposition of uncertainty, as well as the influence of modeling choices related to the approximate posterior $q(\mathcal{W})$. Our method employs an information-theoretic approach (Hüllermeier, 2021), which quantifies aleatoric uncertainty through conditional entropy and epistemic uncertainty using mutual information. Fig. 1 shows the predictive distribution $p(y|x, \mathcal{D})$ and its accompanying uncertainty values obtained by our proposed method for different scenarios of the nuScenes dataset (Caesar et al., 2020). We summarize our contributions as below:

1. We propose a novel method to quantify and decompose the uncertainty of trajectory prediction models, utilizing conditional entropy and mutual information to measure aleatoric and epistemic uncertainty. Both terms are approximated through Monte-Carlo (MC) sampling.

2. We analyze the relationship between uncertainty and prediction error in both in-distribution and out-of-distribution scenarios. Additionally, we evaluate robustness in handling out-of-distribution scenarios.

3. We study how modeling choices of the approximate posterior affect uncertainty calibration and model robustness, considering different configurations of deep ensembles and Dropout.

## 2 Measuring the Uncertainty for Trajectory Prediction

This section details our method for decomposing uncertainty into aleatoric and epistemic components. We start by defining the problem of uncertainty decomposition in trajectory prediction in Sec. 2.1. Then, in Sec. 2.2, we describe our approach for calculating these uncertainties using an MC approximation. We provide pseudo-code in App. A. Finally, we discuss the limitations of our approach with possible avenues to address these limitations in Sec. 2.3.

### 2.1 Problem Statement

Our method focuses on uncertainty quantification in trajectory prediction tasks. The problem is defined as predicting the future trajectory of a target agent in a driving scene based on current observations. Formally, let $x \in \mathbb{R}^{T_{in} \times F_{in}}$ represent the past features of an agent, where $T_{in}$ is the number of observed timesteps and $F_{in}$ denotes the number of input features, such as coordinates, velocities, accelerations, and other relevant data. In line with recent trajectory prediction literature (Deo et al., 2022; Liu et al., 2024; Kim et al., 2021), we also incorporate additional contexts, such as static map information and the past trajectories of surrounding agents, into the model input. A trajectory prediction model $f(x) = y$, parameterized by $\mathcal{W}$, uses this input to estimate a future trajectory $y \in \mathbb{R}^{T_{out} \times F_{out}}$. Here, $T_{out}$ represents the prediction horizon, and $F_{out}$ is the number of output features to predict, such as coordinates. Given the multi-modal nature of an agent's future behavior, an extended version of this model predicts multiple future trajectories. The distribution over potential future outcomes, $p(y|x, \mathcal{W})$, can take various forms, such as a categorical distribution Deo et al. (2022), a mixture of Laplacians (Liu et al., 2024), a GMM (Nayakanti et al., 2023), or a non-parametric form (Jiang et al., 2023). Finally, we define an ensemble (Zhou, 2012) as a set of $M$ trajectory prediction models. These models may have different parameterizations and could belong to different model families. The ensemble can be constructed using various techniques, such as Dropout (Gal & Ghahramani, 2016), Stochastic Gradient Langevin Dynamics (SGLD) (Welling & Teh, 2011), or deep ensembles (Lakshminarayanan et al., 2017). This ensemble introduces a distribution $q(\mathcal{W})$ over neural network parameters, which is an approximation to the true posterior $p(\mathcal{W}|\mathcal{D})$ (Wilson & Izmailov, 2020).

Our objective is to develop a method for uncertainty quantification to assess a model's trustworthiness. However, the type of uncertainty to address is not always clear. On one hand, high uncertainty may stem from novel, previously unseen traffic scenarios. On the other hand, randomness arising from unpredictable driver behavior can lead to multiple plausible predictions. While previous works, such as Gilles et al. (2022) and Janjoš et al. (2023b), do not distinguish between uncertainty types, we argue that decomposing uncertainty is crucial for understanding the sources of a model's predictions. Therefore, following concurrent literature (Der Kiureghian & Ditlevsen, 2009; Hüllermeier, 2021), we decompose uncertainty into epistemic and aleatoric components.

## 2.2 Monte Carlo Approximation of the Conditional Entropy and Mutual Information as a Measure of Aleatoric and Epistemic Uncertainty

To quantify uncertainty, we use entropy as a measure of total uncertainty. This allows us to frame our decomposition in terms of entropy subcomponents. Following Mobiny et al. (2021); Depeweg et al. (2018), we compute epistemic uncertainty as the difference between total and aleatoric uncertainty

$$\underbrace{\mathbf{I}(y, \mathcal{W}|x, \mathcal{D})}_{\text{epistemic uncertainty}} = \underbrace{\mathbf{H}(y|x, \mathcal{D})}_{\text{total uncertainty}} - \underbrace{\mathbb{E}_{p(\mathcal{W}|\mathcal{D})}[\mathbf{H}(y|x, \mathcal{W})]}_{\text{aleatoric uncertainty}}. \tag{2}$$

Above, $\mathbf{I}(y, \mathcal{W}|x, \mathcal{D})$ represents the mutual information between the model's predictions and its parameters, while $\mathbf{H}(y|x, \mathcal{D})$ denotes the total entropy of the predictive distribution. The entropy of a distribution can be computed in closed form for simple cases, such as categorical distributions or univariate Gaussians. However, in trajectory prediction, the predictive distribution can take complex forms, such as a GMM (Nayakanti et al., 2023), making closed-form solutions to Eq. 2 unavailable. To address this, we use a Monte Carlo approximation. For a given input $x$, the entropy is approximated via set of $N$ samples from the predictive distribution, $y_n \sim p(y|x, \mathcal{D})$, as below

$$\mathbf{H}(y|x, \mathcal{D}) = \mathbb{E}_y\left[-\log p(y|x, \mathcal{D})\right] \approx -\frac{1}{N}\sum_{n=1}^{N}\log p(y_n|x, \mathcal{D}) = \hat{\mathbf{H}}(Y|x, \mathcal{D}). \tag{3}$$

Next, we replace the true posterior over neural network parameters $p(\mathcal{W}|\mathcal{D})$ with the approximate posterior $q(\mathcal{W})$. The approximate posterior is a discrete distribution over a set of $M$ neural network parameter values $\mathcal{W}_m$, allowing us to approximate the predictive distribution as

$$p(y|x, \mathcal{D}) = \mathbb{E}_{p(\mathcal{W}|\mathcal{D})}[p(y|x, \mathcal{W})] \approx \mathbb{E}_{q(\mathcal{W})}[p(y|x, \mathcal{W})] = \frac{1}{M}\sum_{m=1}^{M}p(y|x, \mathcal{W}_m). \tag{4}$$

The choice of the model composition $q(\mathcal{W})$ significantly impacts the results, as different models may produce varied predictions, which will be explored further in Sec. 3. We then continue by inserting both Eq. 3 and 4 into the original problem as defined in Eq. 2

$$\mathbf{I}(y, \mathcal{W}|x, \mathcal{D}) \approx \hat{\mathbf{H}}(y|x, \mathcal{D}) - \mathbb{E}_{q(\mathcal{W})}[\hat{\mathbf{H}}(y|x, \mathcal{W})],$$

$$\stackrel{Eq.\ 3}{=} -\frac{1}{N}\sum_{n=1}^{N}\log p(y_n|x, \mathcal{D}) - \mathbb{E}_{q(\mathcal{W})}\left[-\frac{1}{N}\sum_{n=1}^{N}\log p(y_n|x, \mathcal{W})\right],$$

$$\stackrel{Eq.\ 4}{=} -\frac{1}{N}\sum_{n=1}^{N}\log\left(\frac{1}{M}\sum_{m=1}^{M}p(y_n|x, \mathcal{W}_m)\right) + \frac{1}{M}\sum_{m=1}^{M}\frac{1}{N}\sum_{n=1}^{N}\log p(y_n^m|x, \mathcal{W}_m). \tag{5}$$

Above, $y_n^m$ represents the $n$-th sample from the $m$-th model, i.e., $y_n^m \sim p(y|x, \mathcal{W}_m)$. In contrast, $y_n$ represents the $n$-th sample from the predictive distribution after integrating out the weights, i.e., $y_n \sim p(y|x, \mathcal{D})$. We visualize the sampling of $y_n$ in Fig. 2. In essence, we first collect equally-sized sets of $N'$ samples from each distribution $p(y|x, \mathcal{W}_m)$, such that $N = N' \cdot M$. Concatenating them generates $N$ samples from the distribution $p(y|x, \mathcal{D})$, as the weights $\mathcal{W}_m$ are equally weighted.

Our proposed approach formalized in Eq. 2- 5 assumes a generic form of the distribution $p(y|x, \mathcal{W}_m)$. In practice, we use a continuous GMM that is ubiquitous in trajectory prediction for AD, see Sec. 4.1. Thus, we fit samples from a trajectory prediction model to a GMM, or directly use the GMM if the predictor provides one. Details around the GMM design choice can be found in App. B. In Fig. 2, we visualize GMMs fitted to the predictions from $M=3$ ensemble components, as well as samples from each GMM over a two-dimensional grid.

### 2.3 Discussion

The approach presented above effectively quantifies uncertainty in trajectory prediction. Yet, it is important to acknowledge some current limitations and potential solutions. One notable concern is the increased memory and computational burden, which may be prohibitive for a real-time application such as trajectory prediction. A possible avenue to address this limitation is through the use of ensemble distillation. Studies have shown that it is possible to distill an ensemble into a single model, thereby significantly reducing computational overhead while maintaining comparable accuracy Malinin et al. (2019). These technique offers a promising direction for future work, ensuring that our approach remains both efficient and performant.

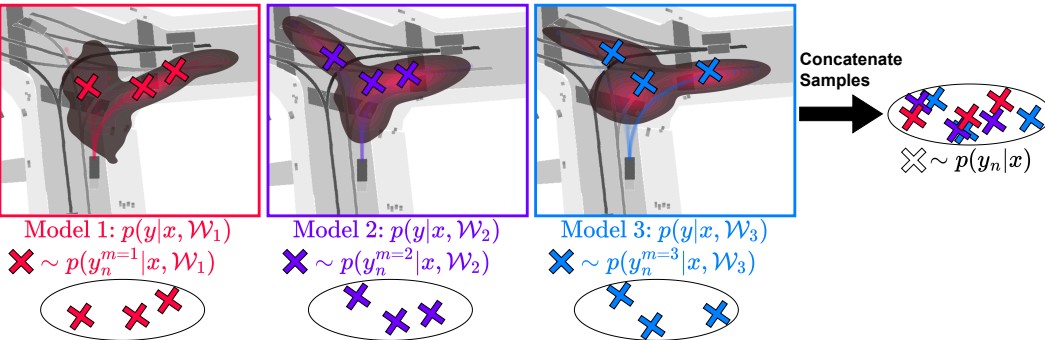

Figure 2: **Generating samples for Monte Carlo approximation.** We fit a GMM to the final positions of trajectories predicted by every member of our ensemble. Then, we sample from each GMM to obtain per-model samples $y_n^m$ for calculating the term of aleatoric uncertainty. Finally, samples originating from all GMMs are aggregated as $y_n$ for calculating the term of total uncertainty.

## 3 EXPERIMENTS

In this paper, we introduce a novel information-theoretic approach to measure and decompose the uncertainty of the predictive distribution of trajectory prediction models in the AD domain. We model the approximate posterior $q(\mathcal{W})$ over neural network weights via sampling-based methods, such as dropout (Gal & Ghahramani, 2016) and deep ensembles Lakshminarayanan et al. (2017). For simplicity, we refer to any collection of neural networks as an ensemble. Our experimental analysis is divided into four parts, where we explore both the uncertainty quantification capabilities of our method and the impact of different ensemble compositions. First, in Sec. 3.1, we benchmark our method against an alternative approach to quantify the uncertainty on the original nuScenes dataset (Caesar et al., 2020), which is a commonly used real-world trajectory prediction dataset for AD. We measure the correlation between the uncertainty and the prediction error and explore how epistemic and aleatoric uncertainties complement each other. In the subsequent parts, we create artificial OOD scenarios by manipulating the nuScenes dataset in various ways. Specifically, we propose four different methods for manipulating the original nuScenes dataset, such as removing lane information or omitting parts of the past trajectories of various agents. A detailed explanation of our nuScenes manipulations is provided in App. D. In the second experimental part in Sec. 3.2, we examine the robustness of various models and ensembles across different OOD scenarios. We observe an overall increase in prediction error, indicating that our artificial OOD scenarios are more challenging than the original dataset. In the third part in Sec. 3.3, we investigate how the correlation between uncertainty and prediction error is affected in these OOD scenarios. Lastly, in Sec. 3.4, we study whether we can detect OOD scenarios by analyzing the different types of uncertainty.

Throughout our experiments, we use our novel method to measure the total uncertainty and decompose it into aleatoric and epistemic components to understand their relative importance. We generate trajectory predictions from the ensemble using the approach described in Distelzweig et al. (2024), which involves Model-Based Risk Minimization (MBRM) to draw trajectories from an ensemble of prediction models. For single models, we generate trajectories via Topk sampling, which selects the most likely trajectories (Liu et al., 2024). We rely on LAformer Liu et al. (2024), PGP Deo et al. (2022), and LaPred Kim et al. (2021) to construct different ensembles of trajectory prediction models. These three models are among the best-performing models with available open-source implementations. In our experiments, we evaluate different ensemble configurations, including deep ensembles, dropout ensembles, and single models. We use an ensemble size of three in all experiments; for deep ensembles, we sample three different models, and for dropout ensembles, we sample three different masks. Prediction performance is assessed in terms of minADE and Minimum Final Displacement Error (minFDE) over $k$ proposals. The minADE$_k$ measures the average point-wise L2 distances between the predicted trajectories and the ground truth, returning the minimum over the $k$ proposals (Caesar et al., 2020). In contrast, minFDE$_k$ considers only the final predicted point. Detailed prediction results for different models and ensemble configurations on the original nuScenes dataset are provided in App. C.

### 3.1 Correlation between Prediction Error and Different Uncertainty Types

Identifying scenarios with high prediction error is critical for safety, as it helps determine when to trust the system or when the driver needs to take control. In this experiment, we study the correlation between different types of uncertainty and prediction error using the original nuScenes dataset. More concretely, we compute the Pearson correlation coefficient $\rho$ between each type of uncertainty and the minADE$_k$. We benchmark our proposed method against Filos et al. (2020), which is an uncertainty quantification approach for planning. Prediction and planning are closely related tasks in AD (Hagedorn et al., 2024), and to the best of our knowledge, Filos et al. (2020) is the only other method with an architecture-agnostic approach that addresses uncertainty quantification in these domains. Filos et al. (2020) estimates the uncertainty by calculating the variance of the log-likelihood of future trajectories with respect to the parameters, i.e., $\text{Var}_{q(\mathcal{W})}[\log p(y|x, \mathcal{W})]$. Unlike our method, this approach requires access to the future trajectory $y$. We report the correlation coefficient between different uncertainty types and the minADE$_5$ in Tab. 1. Additionally, results for minADE$_1$ and minADE$_{10}$ are provided in App. E.

Table 1: Pearson correlation between minADE$_5$ and different uncertainty types on the original nuScenes dataset. We use sampling via MBRM for ensembles and Topk for single models. LP = LaPred (Kim et al., 2021), LF = LAformer (Liu et al., 2024), PGP (Deo et al., 2022), RIP= Robust Imitative Planning (Filos et al., 2020), Dropout (Gal & Ghahramani, 2016).

| | | Deep Ensembles | | | | Dropout Ensembles | | | Single Models | | |
|---|---|---|---|---|---|---|---|---|---|---|---|
| | | 1× LP, LF, PGP | 3× PGP | 3× LF | 3× LP | 3× PGP | 3× LF | 3× LP | 1× PGP | 1× LF | 1× LP |
| Ours | $\rho_{total}$ | 0.38 | 0.35 | 0.39 | 0.27 | 0.31 | 0.37 | 0.21 | 0.27 | 0.26 | 0.15 |
| | $\rho_{aleatoric}$ | 0.36 | 0.34 | 0.38 | 0.19 | 0.31 | 0.36 | 0.15 | 0.27 | 0.26 | 0.15 |
| | $\rho_{epistemic}$ | 0.28 | 0.23 | 0.25 | 0.28 | 0.21 | 0.28 | 0.23 | - | - | - |
| RIP | $\rho_{epistemic}$ | 0.06 | 0.14 | 0.10 | 0.11 | 0.04 | 0.17 | 0.17 | - | - | - |

We first compare the correlation between the minADE$_5$ and different uncertainty types estimated by our method. We observe that for all ensembles except $3 \times$ LP, the total uncertainty has an equal or higher correlation with the prediction error than its individual components, i.e. the aleatoric and epistemic uncertainty. This suggests that both uncertainty sources are complementary. When comparing ensembles against single models, we observe that all ensembles outperform the single models, as these models do not account for epistemic uncertainty. More specifically, when comparing deep ensembles against dropout ensembles, we observe that the former offers a higher correlation with prediction error. This indicates that deep ensembles quantify uncertainty more accurately than dropout, which is in line with the literature on uncertainty quantification with deep ensembles (Lakshminarayanan et al., 2017; Durasov et al., 2021). Lastly, we compare our method against the uncertainty quantification method proposed in Filos et al. (2020), i.e., Robust Imitative Planning (RIP). We observe that our uncertainty quantification method outperforms this approach for all model configurations. This is likely because RIP is based on a heuristic, whereas our method takes a more comprehensive approach. Overall, we observe that the uncertainty estimates obtained by our method provide a useful indication of whether we can trust our model's predictions or not.

### 3.2 Robustness of Predictions in OOD Scenarios

In the previous experiment, we analyzed the correlation between uncertainty and prediction error in in-distribution scenarios. We now shift our focus to examining whether prediction performance degrades in OOD scenarios and to what extent. We report the changes in the minADE$_5$ metric with respect to the original dataset in Fig. 3. App. I and App. J provide additional visualizations for minADE$_1$ and minADE$_{10}$ as well as numerical values.

Overall, we observe that prediction error increases across all datasets in OOD scenarios. However, model ensembles consistently outperform individual models as in all model configurations, more than 50% of the data points fall within the upper green triangle on Fig. 3. This suggests that ensembles offer greater robustness and resilience in OOD scenarios. When comparing deep ensembles composed of the same model to their dropout-based alternatives, the performance is similar

when considering the fraction of data points in the green triangle. For instance, the dropout ensemble performs better for PGP, while for LaPred, both the dropout and deep ensemble configurations show equal performance. In contrast, LAformer shows better performance with deep ensembles. However, when examining the mixed deep ensemble that combines different models, we observe a significant increase in performance with all data points lying within the green triangle.

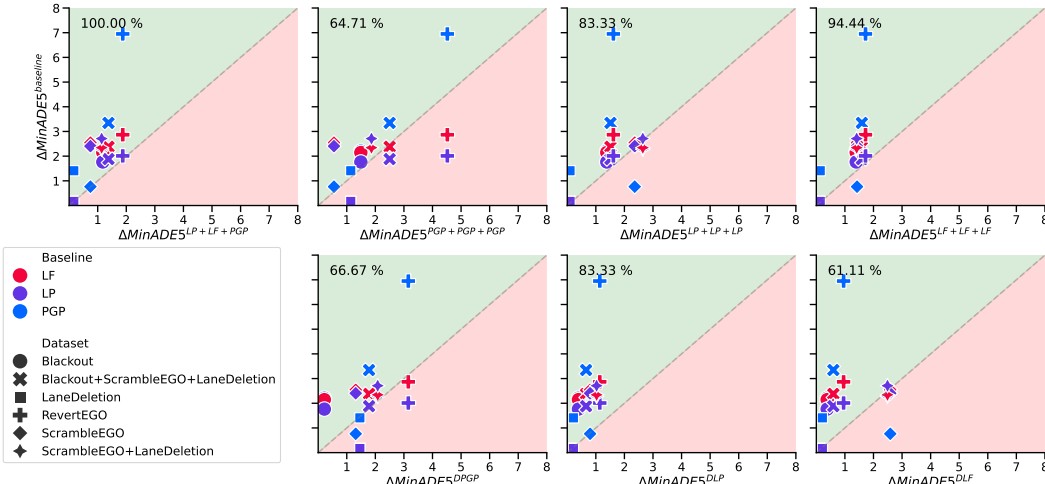

Figure 3: Differences ($\Delta$) in MinADE$_5$ between the original dataset and the corresponding out-of-distribution dataset for baseline models (y-axis) and ensembles (x-axis). Different colors correspond to various baseline models, while different markers denote distinct datasets. Markers positioned in the red area (lower triangle) of each plot indicate that the ensemble exhibits a larger $\Delta$MinADE$_5$ compared to the baseline. Conversely, markers in the green area signify a smaller $\Delta$MinADE$_5$ for the ensemble. Percentages indicate how often the ensemble outperforms the baseline.

### 3.3 QUANTIFYING THE UNCERTAINTY IN OOD SCENARIOS

In Sec. 3.1, we investigated whether the uncertainty estimates from our method offer indications of the reliability of our model's predictions. However, it remains unclear if these findings are also applicable to OOD scenarios. In this experiment, we analyze the correlation between uncertainty and prediction error in OOD scenarios across different ensembles, and we compare these correlation coefficients with those obtained from the original dataset. We report the correlation coefficient between the total uncertainty and the minADE$_5$ in Fig. 4. App. G and App. H provide additional visualizations for minADE$_1$ and minADE$_{10}$ as well as numerical results.

We first compare the correlation values from the original dataset represented by the circle marker in Fig. 4 with those from the OOD datasets represented by all other markers. The results present a mixed picture – in some OOD scenarios, the correlation coefficient decreases while in others, it increases. Nevertheless, there is a general trend toward a decrease in the correlation coefficient in most OOD cases. Interesting observations can be made when focusing on a specific model ensemble, e.g. the mixed ensemble consisting of LAformer, PGP, and LaPred. Despite being applied to OOD scenarios, this ensemble maintains a higher correlation coefficient than that achieved by the alternative uncertainty quantification method proposed by Filos et al. (2020) on the in-distribution scenarios of the original dataset. This suggests that our proposed method and selected ensemble models offer more robust uncertainty quantification even under challenging conditions.

Next, we investigate whether using an ensemble of models is more advantageous than relying on a single model in OOD scenarios. To assess this, we compare the correlation between uncertainty and prediction error for ensembles versus individual models. Our findings reveal that the ensemble configurations consistently outperform the single model baselines. This conclusion is supported by the fact that in every configuration, more than 50% of the data points lie within the green triangle. This suggests that ensembles provide a more reliable measure of uncertainty in OOD scenarios compared to single models. Lastly, we compare different model ensembles. Specifically, we compare

deep ensembles composed of the same model versus their dropout-based alternatives. In two out of three cases, dropout ensembles outperform deep ensembles in terms of the number of data points within the green triangle. However, when we consider a mixed deep ensemble, which combines different models rather than ensembling multiple instances of the same model, we observe a notable improvement in performance. The number of data points within the green triangle increases, signifying that the mixed ensemble achieves a higher correlation between uncertainty and prediction error. This suggests that mixed ensembles, which benefit from a high model diversity, provide better uncertainty quantification than deep ensembles composed of a single model type. This conclusion aligns with our earlier findings, where mixed ensembles consistently performed the best or matched other ensemble configurations in terms of robustness in OOD scenarios. Therefore, we conclude that mixed deep ensembles are the most effective choice for handling OOD scenarios.

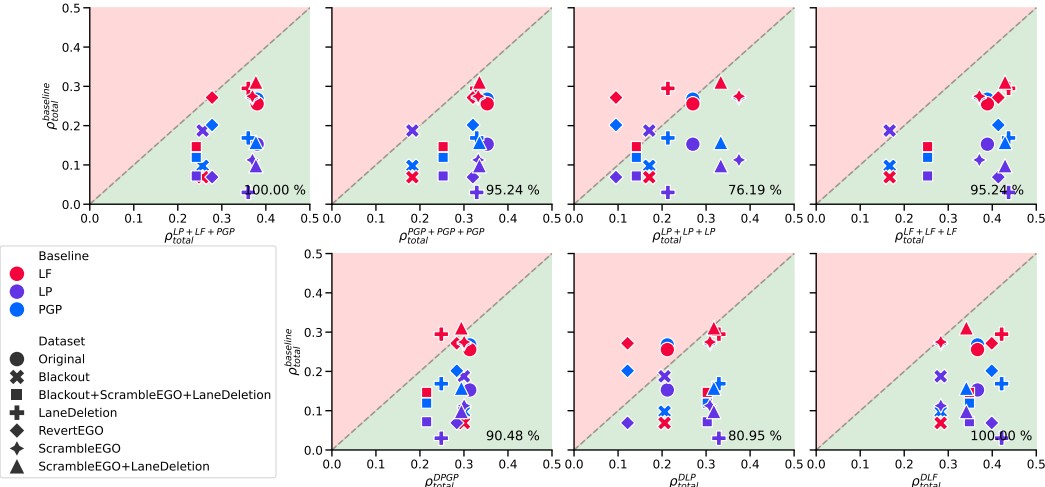

Figure 4: Pearson correlation coefficient $\rho$ between total uncertainty and MinADE$_5$ for baseline models ($y$-axis) and ensembles ($x$-axis) over the validation set. Different colors represent various baseline models, while different markers indicate distinct datasets. Markers located in the red area (upper triangle) of each plot signify that the ensemble shows a lower correlation $\rho_{total}$ compared to the baseline. Conversely, markers in the green area (lower triangle) indicate a higher correlation for the ensemble. The numerical value in the bottom right corner of each plot represents the fraction of data points that fall within the green area.

## 3.4 Detecting OOD Scenarios

In this experiment, our objective is to determine whether we can identify OOD scenarios in the first place. Recognizing such scenarios is critical for improving the performance and robustness of an AD system over time, as it facilitates the collection of challenging cases for re-training and evaluation. We present the uncertainty values for different types of uncertainty in Fig. 5 for both the original nuScenes dataset and various OOD scenarios. For this analysis, we restrict our focus to a mixed deep ensemble consisting of LAformer, PGP, and LaPred, as this ensemble demonstrated the best correlation between uncertainty and prediction error in previous experiments. Additionally, we include OOD detection results for other model ensembles in App. F.

When analyzing epistemic uncertainty, we observe that OOD scenarios exhibit a higher median value than the upper quartile of the original dataset, with the exception of the blackout scenario, where only the median of the original dataset is exceeded. In terms of aleatoric uncertainty, the median for OOD scenarios consistently exceeds the median observed in the original dataset. The total uncertainty follows a similar pattern to aleatoric uncertainty but exhibits a more pronounced difference between OOD and in-distribution cases. These trends indicate that OOD scenarios can be identified with the highest confidence by assessing epistemic uncertainty, a finding that aligns with existing research in uncertainty quantification (Hüllermeier, 2021).

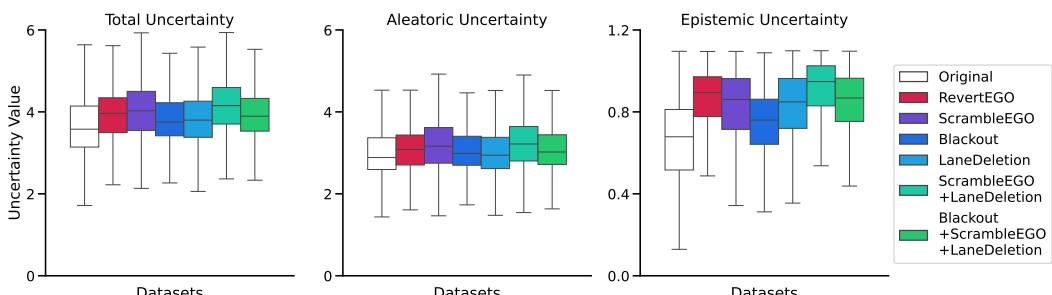

Figure 5: Total, aleatoric, and epistemic uncertainties for a mixed ensemble ($1 \times$ LP, LF, PGP) for the original dataset as well as all out-of-distribution datasets.

## 4 RELATED WORK

Anticipating the future motion of traffic participants is a critical component of autonomous driving systems (Hu et al., 2023). Due to the safety-critical nature of these systems, it's essential to account for and propagate uncertainties across the entire prediction stack (McAllister et al., 2017). For instance, planners need to factor in motion prediction uncertainty to accurately assess the risks associated with various driving maneuvers (Filos et al., 2020). In the following subsections, we review related work on both motion prediction as well as quantification and decomposition of uncertainty.

### 4.1 MOTION PREDICTION FOR AUTONOMOUS DRIVING

The future motion of other traffic participants is influenced by a multitude of observable and unobservable factors, rendering it a challenging modeling task. These factors include, among others, the latent goals and preferences of traffic participants, social norms and traffic rules, complex interactions with surrounding traffic, as well as constraints induced by the static environment (Rudenko et al., 2020). The shortcomings of the perception system, which provides noisy and partial observations, pose an additional challenge. These challenges necessitate a probabilistic formulation of the prediction task to adequately model the uncertain and multi-modal nature of future motion. In general, prediction models consist of two components: a behavior backbone, which encodes the traffic scene, and a decoder, which models the predictive distribution. We will highlight various implementations of the two components below.

Early prediction approaches (Casas et al., 2018; Chai et al., 2020; Phan-Minh et al., 2020) propose encoding the past trajectory of observed traffic participants and the elements of the static environment (e.g., lane boundaries, crosswalks, traffic signs) by rendering the scene in a semantic bird's eye view image and applying well-established convolutional neural networks (He et al., 2016). Such image-based representations of the scene have largely been replaced by vectorized representations due to their inefficiency (Gao et al., 2020; Zhao et al., 2021; Kim et al., 2021; Deo et al., 2022; Nayakanti et al., 2023). In a vectorized representation, all entities of the static and dynamic environment are approximated by a sequence of vectors. Models for sequential data, such as temporal convolutional networks (van den Oord et al., 2016) or recurrent neural networks (Hochreiter & Schmidhuber, 1997; Chung et al., 2014) are used to encode the sequences and interactions between entities are modeled using pooling operations (Alahi et al., 2016), graph neural networks (Hamilton, 2020), or Transformers (Vaswani et al., 2017).

The future motion of traffic participants is typically characterized by a sequence of states over multiple time steps, known as trajectories (Ngiam et al., 2022; Varadarajan et al., 2022). Several strategies are employed to capture the highly multi-modal distribution over trajectories conditioned on the encoded scene. Many approaches represent the distribution by a set of trajectories with associated mode probabilities. The trajectories are either regressed by the model (Cui et al., 2019; Liang et al., 2020; Kim et al., 2021; Deo et al., 2022) or fixed a priori (Phan-Minh et al., 2020). Other approaches use parametric mixture distributions, such as GMMs (Khandelwal et al., 2020; Tolstaya et al., 2021; Varadarajan et al., 2022) or mixtures of Laplacians (Liu et al., 2024). Alternatively, generative models such as conditional variational autoencoders (Lee et al., 2017; Bhattacharyya et al., 2019;

Salzmann et al., 2020; Janjoš et al., 2023a), generative adversarial networks (Gupta et al., 2018; Huang et al., 2020; Gómez-Huélamo et al., 2022), normalizing flows (Schöller & Knoll, 2021), or diffusion models (Jiang et al., 2023) model the trajectory distribution via latent variables.

## 4.2 UNCERTAINTY MODELING, DECOMPOSITION AND QUANTIFICATION

The majority of current trajectory prediction models solely account for aleatoric uncertainty by modeling a probability distribution on the output space (Varadarajan et al., 2022). To incorporate epistemic uncertainty in a theoretically sound manner, one can adopt a Bayesian framework (Kendall & Gal, 2017; Depeweg et al., 2018; Wilson & Izmailov, 2020; Wilson, 2020). A Bayesian neural network assumes a distribution over the network weights instead of a point estimate to account for the lack of knowledge about the data-generating process (Hüllermeier, 2021; Jospin et al., 2022). Since analytically evaluating the posterior distribution over the weights is intractable for modern neural networks, approximate inference techniques such as Variational Inference (VI) or forms of MCMC must be considered (Jospin et al., 2022). Due to its simplicity, MC Dropout, which can be interpreted as an approximate VI method (Gal & Ghahramani, 2016), is used by many perception approaches in AD (Kendall & Gal, 2017; Abdar et al., 2021) and is also employed by Janjoš et al. (2023b) for modeling epistemic uncertainty of a trajectory predictor. Another well-established approach to account for epistemic uncertainty is deep ensembles (Lakshminarayanan et al., 2017; Jospin et al., 2022; Wilson & Izmailov, 2020). Prior work (Filos et al., 2020) uses deep ensembles to approximate the posterior distribution in their epistemic uncertainty-aware planning method. We apply MC Dropout as well as deep ensembles to approximate the uncertainty over network weights and systematically assess their performance in the context of trajectory prediction.

A common information-theoretical measure for the uncertainty is the entropy of the predictive distribution as a measure of the total uncertainty, which can be additively decomposed into the conditional entropy and mutual information, representing a measure of aleatoric and epistemic uncertainty (Depeweg et al., 2018; Smith & Gal, 2018; Hüllermeier, 2021; Wimmer et al., 2023). Alternative measures based on the variance were proposed by Depeweg et al. (2018). While variance-based measures are suitable in cases where the predictive distribution is a uni-modal Gaussian, it is less suitable for multi-modal outputs, such as trajectories. Our approach thus relies on entropy-based measures to quantify the uncertainties of trajectory prediction models. However, variance can be useful in other contexts; e.g., Gilles et al. (2022) uses the variance of the predicted heat map over future positions as an uncertainty measure. Another variance-based uncertainty heuristic is proposed by Filos et al. (2020) in the related field of motion planning for AD. This approach, however, only quantifies the epistemic uncertainty and requires access to the future trajectory. Some methods train separate models (Pustynnikov & Eremeev, 2021) or include additional heads with auxiliary tasks (Wiederer et al., 2023; Janjoš et al., 2023b) to learn a proxy measure of the uncertainty of a trajectory prediction model without a proper decomposition.

To the best of our knowledge, we are the first to thoroughly investigate the theoretically sound modeling, decomposition, and quantification of uncertainties for trajectory prediction models. All other published trajectory prediction approaches either do not holistically consider all three aspects or resort to heuristics.

## 5 CONCLUSION

Understanding and addressing uncertainty in probabilistic motion prediction for AD remains a key challenge. This paper addresses this gap by proposing a general approach to quantify and decompose uncertainty using an information-theoretic framework. We demonstrate that our estimates of aleatoric and epistemic uncertainty provide meaningful indicators of prediction error, making them reliable for assessing prediction performance. Through an extensive evaluation, we examine both in-distribution and out-of-distribution scenarios under various posterior assumptions. Overall, our approach advances principled uncertainty modeling in motion prediction for AD.

A promising future direction is to integrate our uncertainty quantification framework in the planning system of an autonomous vehicle (Teng et al., 2023; Hagedorn et al., 2024). This opens the possibility of combining autonomous driving planning research with risk-sensitive reinforcement learning (Depeweg et al., 2018), enabling the system to make informed decisions in uncertain situations.

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

## A  PSEUDO CODE

The below pseudo code outlines the method of computing and decomposing uncertainties. Here, $\mathcal{G}_m$ refers to the Gaussian Mixture Model (GMM) from which we draw and score samples. We use the functions `Sample` and `Score` that implement the standard procedures for sampling and computing the probability of a sample given the distribution (Dempster et al., 1977). The `Sample` function takes a probability distribution as input and returns a sample. The `Score` function takes both the probability distribution and a sample as inputs and returns the likelihood (or score) of the sample.

---

**Algorithm 1** Monte-Carlo approximation of Uncertainty Types

---

 **Input:** $\{f_1, ..., f_M\}$                ▷ Set of models
    $x$                      ▷ Input

 **Output:** $\hat{\mathbf{I}}(y, \mathcal{W}|x, \mathcal{D})$             ▷ Epistemic uncertainty
     $\mathbb{E}_{q(\mathcal{W})}[\hat{\mathbf{H}}(y|x, \mathcal{W})]$         ▷ Aleatoric uncertainty
     $\hat{\mathbf{H}}[y|x, \mathcal{D}]$              ▷ Total uncertainty

1: **for** $m \leftarrow 1$ to $M$ **do**
2:   $\mathcal{G}_m \leftarrow \text{GMM}(f_m(x))$             ▷ Fit a GMM to $f_m$
3: **end for**

 # Aleatoric uncertainty
4: **for** $m \leftarrow 1$ to $M$ **do**
5:   **for** $n \leftarrow 1$ to $N$ **do**
6:     $y_n^m \leftarrow \text{Sample}(\mathcal{G}_m)$
7:     $p(y_n^m|x, \mathcal{W}_m) \leftarrow \text{Score}(\mathcal{G}_m, y_n^m)$
8:   **end for**
9:   $\hat{\mathbf{H}}(y|x, \mathcal{W}_m) \leftarrow -\frac{1}{N}\sum_{n=1}^{N} \log p(y_n^m|x, \mathcal{W}_m)$       ▷ Eq. 3
10: **end for**
11: $\mathbb{E}_{q(\mathcal{W})}[\hat{\mathbf{H}}(y|x, \mathcal{W})] \leftarrow \frac{1}{M}\sum_{m=1}^{M} \hat{\mathbf{H}}(y|x, \mathcal{W}_m)$       ▷ Eq. 4

 # Total uncertainty
12: **for** $m \leftarrow 1$ to $M$ **do**        ▷ Generate $N'$ sample from each model
13:   **for** $n \leftarrow (m-1)N'$ to $mN'$ **do**
14:     $y_n \leftarrow \text{Sample}(\mathcal{G}_m)$
15:   **end for**
16: **end for**
17: **for** $n \leftarrow 1$ to $N$ **do**
18:   **for** $m \leftarrow 1$ to $M$ **do**
19:     $p(y_n|x, \mathcal{W}_m) \leftarrow \text{Score}(\mathcal{G}_m, y_n)$
20:   **end for**
21:   $p(y_n|x, \mathcal{D}) \leftarrow \frac{1}{M}\sum_{m=1}^{M} p(y_n|x, \mathcal{W}_m)$        ▷ Eq. 4
22: **end for**
23: $\hat{\mathbf{H}}(y|x, \mathcal{D}) \leftarrow -\frac{1}{N}\sum_{n=1}^{N} \log p(y_n|x, \mathcal{D})$        ▷ Eq. 3

 # Epistemic uncertainty
24: $\hat{\mathbf{I}}(y, \mathcal{W}|x, \mathcal{D}) \leftarrow \hat{\mathbf{H}}(y|x, \mathcal{D}) - \mathbb{E}_{q(\mathcal{W})}[\hat{\mathbf{H}}(y|x, \mathcal{W})]$      ▷ Eq. 2

---

## B  DISTRIBUTION ASSUMPTIONS

In this section, we briefly cover the choice of a GMM for the distribution $p(y|x, \mathcal{W}_m)$ assumed in Eq. 5. Note that $p(y|x, \mathcal{W}_m)$ can initially take either categorical or continuous forms, depending on the model. However, we transform $p(y|x, \mathcal{W}_m)$ into a GMM, a continuous distribution, for two reasons. First, for a categorical distribution, the probability $p(y|x, \mathcal{W}_m)$ is zero for any sample not generated by the model with parameters $\mathcal{W}_m$. This is because categorical distributions only assign non-zero probabilities to discrete outcomes they were trained on, making them unsuitable for

continuous trajectory prediction tasks. A GMM, on the other hand, smooths out the discrete distribution, which aligns better with the continuous nature of the regression task. Second, handling a mixture of categorical and continuous distributions introduces numerical issues. When $p(y|x, \mathcal{W}_m)$ is continuous, we model the logarithm of the density. However, when it is discrete, we model the probability directly. This discrepancy leads to numerical challenges during the log-sum-exp operation when calculating $\hat{\mathbf{H}}(y|x, \mathcal{D})$. The resulting predictive distribution $p(y|x, \mathcal{D})$ is then represented as a mixture of $M$ GMMs, where each GMM corresponds to one of the models in the ensemble. For computational convenience, we perform these calculations only for the final predicted timestep, i.e., the endpoint. Fig. 2 shows the GMM fitted to the final positions of the predictions from each model, which represents the distribution of the final coordinates of the predicted trajectories over a two-dimensional continuous grid.

## C  PERFORMANCE ON THE ORIGINAL NUSCENES DATASET

In the below table, we report the performance of different ensembles and base models on the original nuScenes dataset. We generate trajectory predictions from the ensemble using the approach described in Distelzweig et al. (2024), which involves MBRM to draw trajectories from an ensemble of prediction models. For single models, we generate trajectories via Topk sampling, which selects the most likely trajectories (Liu et al., 2024).

Table 2: minADE and minFDE for different $k$ values and ensembling strategies. LP = LaPred (Kim et al., 2021), LF = LAformer (Liu et al., 2024), PGP (Deo et al., 2022).

| Model(s) | minADE (↓) | | | minFDE (↓) | | |
|---|---|---|---|---|---|---|
| | $k = 1$ | $k = 5$ | $k = 10$ | $k = 1$ | $k = 5$ | $k = 10$ |
| $1 \times \{$LP, LF, PGP$\}$ | 2.89 | 1.22 | 0.93 | 6.70 | 2.37 | 1.54 |
| $3 \times$LP | 3.08 | 1.34 | 1.08 | 7.16 | 2.66 | 1.88 |
| $3 \times$LF | 2.82 | 1.20 | 0.92 | 6.52 | 2.34 | 1.54 |
| $3 \times$PGP | 2.97 | 1.22 | 0.94 | 6.87 | 2.37 | 1.55 |
| Dropout + LP | 3.15 | 1.39 | 1.14 | 7.31 | 2.78 | 2.01 |
| Dropout + LF | 2.94 | 1.28 | 1.00 | 6.86 | 2.53 | 1.72 |
| Dropout + PGP | 3.07 | 1.26 | 0.97 | 7.09 | 2.46 | 1.64 |
| LP | 3.56 | 1.53 | 1.17 | 8.48 | 3.13 | 2.18 |
| LF | 3.07 | 1.51 | 0.95 | 7.12 | 3.06 | 1.61 |
| PGP | 3.17 | 1.28 | 0.96 | 7.38 | 2.49 | 1.57 |

In MBRM, the below equation is optimized during inference with respect to the trajectories $\tilde{y}$

$$\hat{y} \approx \underset{\tilde{y}}{\operatorname{argmin}} \sum_{m=1}^{M} \sum_{n=1}^{N} \frac{w_n^m}{M} \text{minADE}_k(y_n^m, \tilde{y}). \tag{6}$$

Above $y_n^m \in \mathbb{R}^{T \times 2}$ represents the $n$-th proposal trajectory of the $m$-th model and $w_n^m \in \mathbb{R}^+$ represents the corresponding weight. The number of total models is $M$. For more details, we refer to Distelzweig et al. (2024).

## D  GENERATION OF OOD SCENARIOS

We create artificial OOD scenarios by manipulating the original nuScenes dataset. Below we describe our manipulation techniques.

- RevertEGO: We revert the history of the ego/target vehicle.
- ScrambleEGO: We randomly shuffle the history of the ego/target vehicle.
- Blackout: We set 1/2 of the history to zero for the ego/target and all surrounding vehicles.
- LaneDeletion: We randomly delete 3/4 of all lanes considered by the model.

Beyond that, we consider combinations of manipulations.

# E ADDITIONAL RESULTS ON THE ORIGINAL NUSCENES DATASET

Table 3: Pearson correlation between minADE$_1$ and different uncertainty types on the original nuScenes dataset. We use sampling via MBRM for ensembles and Topk for single models. LP = LaPred (Kim et al., 2021), LF = LAformer (Liu et al., 2024), PGP (Deo et al., 2022), RIP= Robust Imitative Planning (Filos et al., 2020), Dropout (Gal & Ghahramani, 2016).

| | | Deep Ensembles | | | | Dropout Ensembles | | | Single Models | | |
|---|---|---|---|---|---|---|---|---|---|---|---|
| | | 1× LP, LF, PGP | 3× PGP | 3× LF | 3× LP | 3× PGP | 3× LF | 3× LP | 1× PGP | 1× LF | 1× LP |
| Ours | $\rho_{total}$ | 0.28 | 0.29 | 0.25 | 0.22 | 0.26 | 0.24 | 0.17 | 0.17 | 0.22 | 0.16 |
| | $\rho_{aleatoric}$ | 0.28 | 0.27 | 0.24 | 0.17 | 0.26 | 0.24 | 0.14 | 0.17 | 0.22 | 0.16 |
| | $\rho_{epistemic}$ | 0.17 | 0.19 | 0.18 | 0.18 | 0.16 | 0.18 | 0.09 | - | - | - |
| RIP | $\rho_{epistemic}$ | 0.06 | 0.06 | 0.05 | 0.08 | 0.05 | 0.05 | 0.05 | - | - | - |

Table 4: Pearson correlation between minADE$_{10}$ and different uncertainty types on the original nuScenes dataset. We use sampling via MBRM for ensembles and Topk for single models. LP = LaPred (Kim et al., 2021), LF = LAformer (Liu et al., 2024), PGP (Deo et al., 2022), RIP= Robust Imitative Planning (Filos et al., 2020), Dropout (Gal & Ghahramani, 2016).

| | | Deep Ensembles | | | | Dropout Ensembles | | | Single Models | | |
|---|---|---|---|---|---|---|---|---|---|---|---|
| | | 1× LP, LF, PGP | 3× PGP | 3× LF | 3× LP | 3× PGP | 3× LF | 3× LP | 1× PGP | 1× LF | 1× LP |
| Ours | $\rho_{total}$ | 0.38 | 0.33 | 0.39 | 0.25 | 0.30 | 0.37 | 0.19 | 0.29 | 0.39 | 0.12 |
| | $\rho_{aleatoric}$ | 0.36 | 0.31 | 0.38 | 0.16 | 0.30 | 0.36 | 0.12 | 0.29 | 0.39 | 0.12 |
| | $\rho_{epistemic}$ | 0.28 | 0.24 | 0.24 | 0.30 | 0.20 | 0.29 | 0.25 | - | - | - |
| RIP | $\rho_{epistemic}$ | 0.08 | 0.17 | 0.11 | 0.12 | 0.06 | 0.21 | 0.19 | - | - | - |

# F ADDITIONAL PLOTS FOR OOD DETECTION

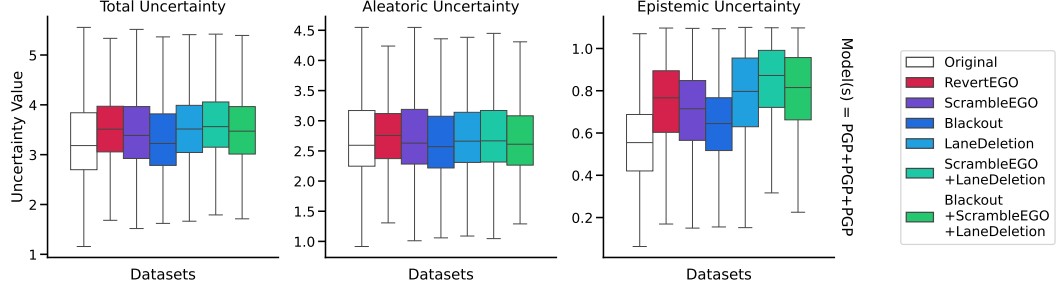

Figure 6: Total, aleatoric, and epistemic uncertainties for a PGP ensemble (3 × PGP) for the original dataset as well as all out-of-distribution datasets.

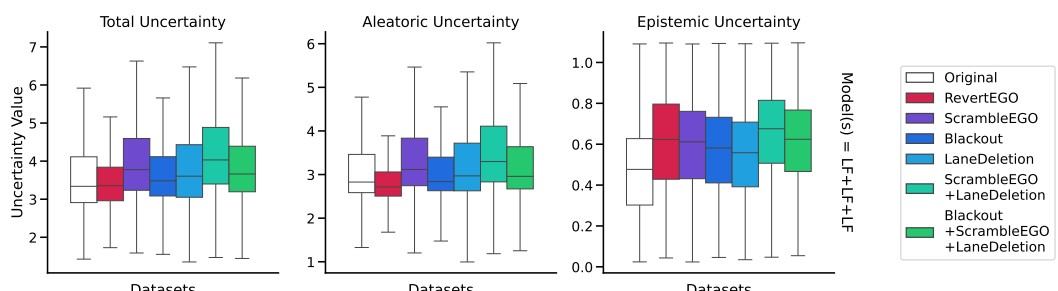

Figure 7: Total, aleatoric, and epistemic uncertainties for a LAformer ensemble (3 × LF) for the original dataset as well as all out-of-distribution datasets.

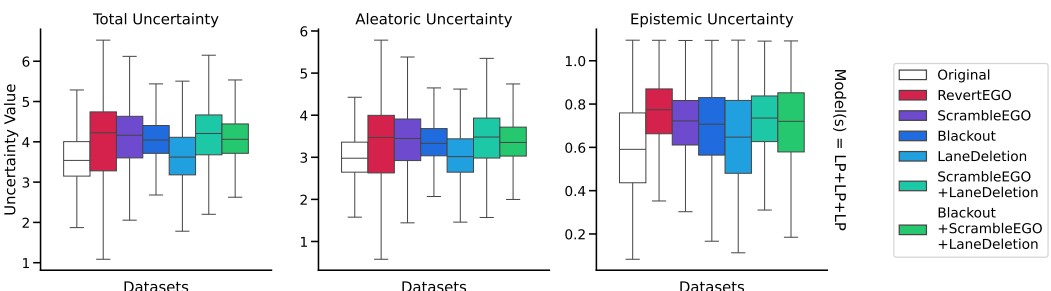

Figure 8: Total, aleatoric, and epistemic uncertainties for a LaPred ensemble (3 × LP) for the original dataset as well as all out-of-distribution datasets.

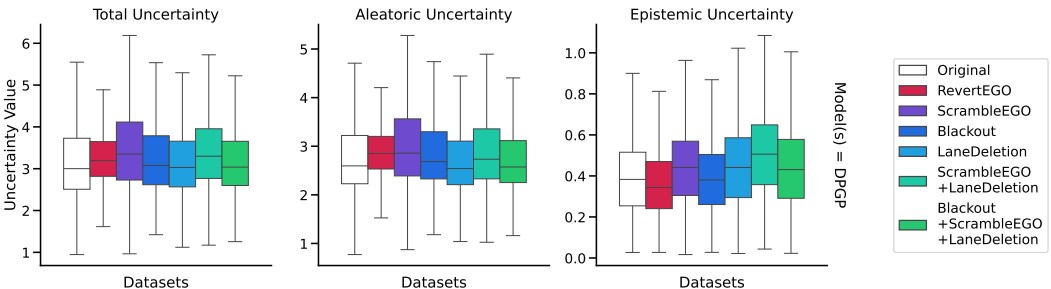

Figure 9: Total, aleatoric, and epistemic uncertainties for Dropout + PGP for the original dataset as well as all out-of-distribution datasets.

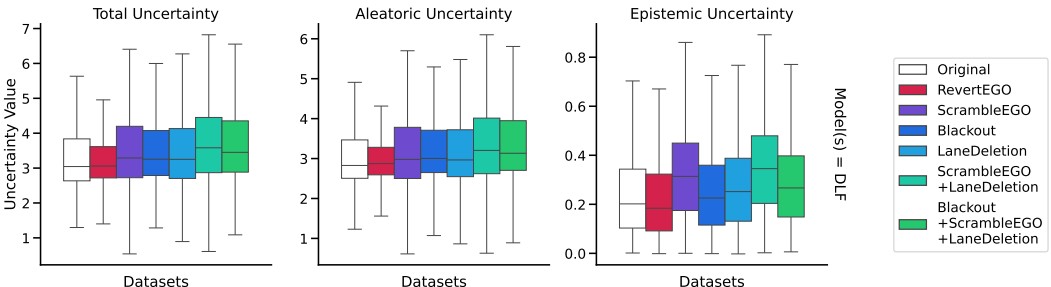

Figure 10: Total, aleatoric, and epistemic uncertainties for Dropout + LAformer for the original dataset as well as all out-of-distribution datasets.

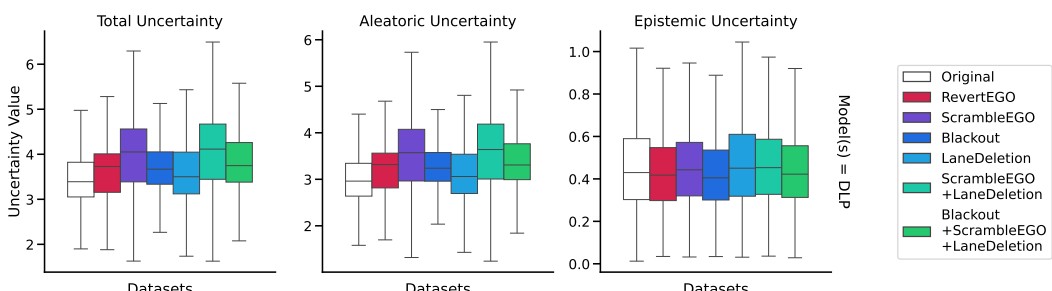

Figure 11: Total, aleatoric, and epistemic uncertainties for Dropout + LaPred for the original dataset as well as all out-of-distribution datasets.

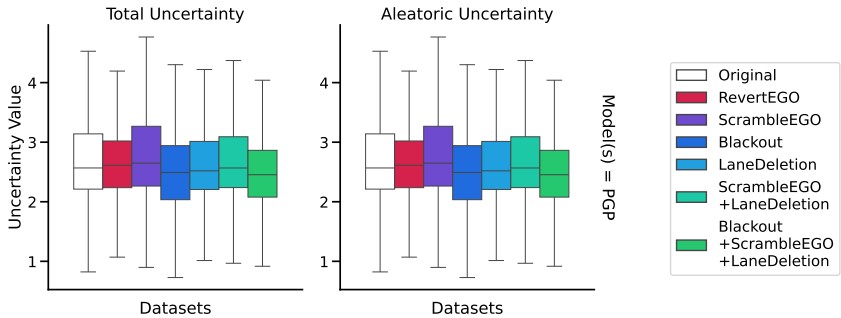

Figure 12: Total and aleatoric uncertainties for PGP for the original dataset as well as all out-of-distribution datasets.

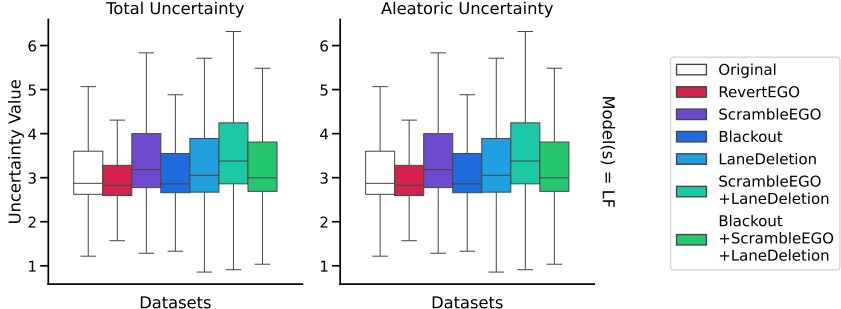

Figure 13: Total and aleatoric uncertainties for LAformer for the original dataset as well as all out-of-distribution datasets.

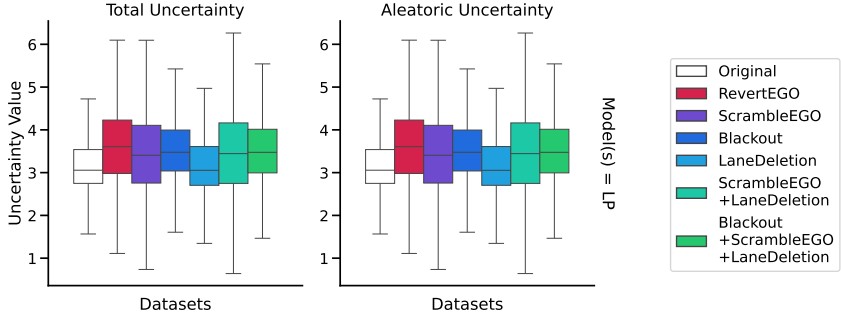

Figure 14: Total and aleatoric uncertainties for LaPred for the original dataset as well as all out-of-distribution datasets.

# G   ADDITIONAL CORRELATION PLOTS FOR OOD SCENARIOS

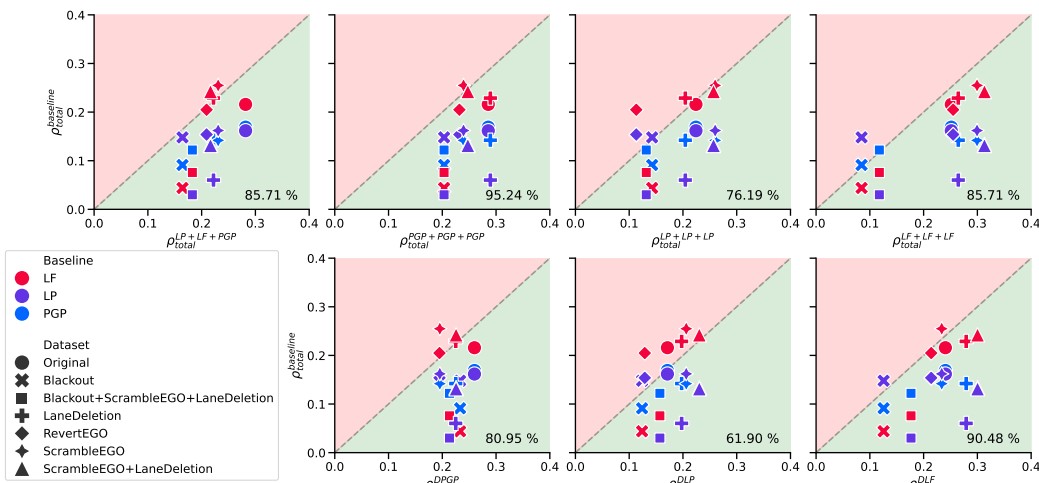

Figure 15: Pearson correlation coefficient $\rho$ between total uncertainty and MinADE$_1$ for baseline models ($y$-axis) and ensembles ($x$-axis) over the validation set. Different colors represent various baseline models, while different markers indicate distinct datasets.

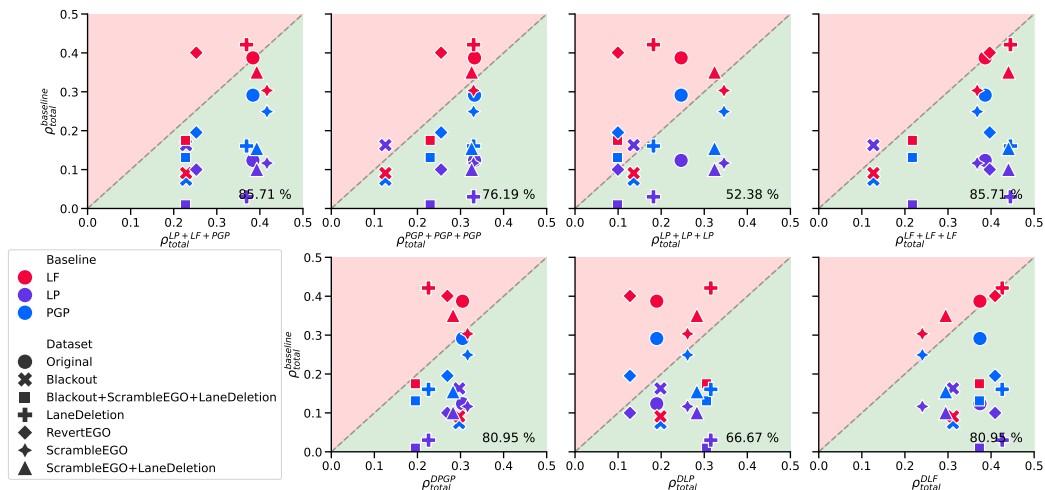

Figure 16: Pearson correlation coefficient $\rho$ between total uncertainty and MinADE$_{10}$ for baseline models ($y$-axis) and ensembles ($x$-axis) over the validation set. Different colors represent various baseline models, while different markers indicate distinct datasets.

## H  VALUES OF THE CORRELATION COEFFICIENT

Table 5: Pearson correlation coefficient between the different types of uncertainties (total, aleatoric, and epistemic) estimated by our approach and MinADE$_1$ across all out-of-distribution and the original dataset and all ensembles as well as single models.

| Model(s) | DS | Ours | | | Baseline |
|---|---|---|---|---|---|
| | | $\rho_{total}$ | $\rho_{aleatoric}$ | $\rho_{epistemic}$ | |
| $1 \times \{$LP, LF, PGP$\}$ | Original | 0.28 | 0.28 | 0.17 | 0.06 |
| | Blackout | 0.16 | 0.14 | 0.15 | 0.06 |
| | Blackout+ScrambleEGO+LaneDeletion | 0.18 | 0.15 | 0.16 | 0.09 |
| | LaneDeletion | 0.22 | 0.18 | 0.18 | 0.08 |
| | RevertEGO | 0.21 | 0.15 | 0.29 | 0.07 |
| | ScrambleEGO | 0.23 | 0.22 | 0.15 | 0.05 |
| | ScrambleEGO+LaneDeletion | 0.22 | 0.19 | 0.17 | 0.08 |
| $3 \times$ PGP | Original | 0.29 | 0.27 | 0.19 | 0.06 |
| | Blackout | 0.20 | 0.19 | 0.14 | 0.05 |
| | Blackout+ScrambleEGO+LaneDeletion | 0.20 | 0.18 | 0.15 | 0.12 |
| | LaneDeletion | 0.29 | 0.25 | 0.21 | 0.14 |
| | RevertEGO | 0.23 | 0.20 | 0.24 | 0.06 |
| | ScrambleEGO | 0.24 | 0.24 | 0.21 | 0.06 |
| | ScrambleEGO+LaneDeletion | 0.25 | 0.25 | 0.21 | 0.17 |
| $3 \times$ LP | Original | 0.22 | 0.17 | 0.18 | 0.08 |
| | Blackout | 0.14 | 0.13 | 0.07 | 0.06 |
| | Blackout+ScrambleEGO+LaneDeletion | 0.13 | 0.11 | 0.08 | 0.09 |
| | LaneDeletion | 0.20 | 0.15 | 0.17 | 0.07 |
| | RevertEGO | 0.11 | 0.06 | 0.26 | 0.07 |
| | ScrambleEGO | 0.26 | 0.19 | 0.28 | 0.04 |
| | ScrambleEGO+LaneDeletion | 0.26 | 0.19 | 0.28 | 0.01 |
| $3 \times$ LF | Original | 0.25 | 0.24 | 0.18 | 0.05 |
| | Blackout | 0.08 | 0.07 | 0.10 | 0.05 |
| | Blackout+ScrambleEGO+LaneDeletion | 0.12 | 0.10 | 0.12 | 0.05 |
| | LaneDeletion | 0.26 | 0.26 | 0.16 | 0.06 |
| | RevertEGO | 0.25 | 0.23 | 0.19 | 0.09 |
| | ScrambleEGO | 0.30 | 0.28 | 0.27 | 0.04 |
| | ScrambleEGO+LaneDeletion | 0.31 | 0.30 | 0.24 | 0.05 |
| Dropout + PGP | Original | 0.26 | 0.26 | 0.16 | 0.05 |
| | Blackout | 0.23 | 0.23 | 0.15 | 0.05 |
| | Blackout+ScrambleEGO+LaneDeletion | 0.21 | 0.20 | 0.16 | 0.09 |
| | LaneDeletion | 0.22 | 0.21 | 0.17 | 0.11 |
| | RevertEGO | 0.19 | 0.19 | 0.12 | 0.08 |
| | ScrambleEGO | 0.20 | 0.19 | 0.14 | 0.03 |
| | ScrambleEGO+LaneDeletion | 0.23 | 0.20 | 0.20 | 0.09 |
| Dropout + LP | Original | 0.17 | 0.14 | 0.09 | 0.05 |
| | Blackout | 0.12 | 0.11 | 0.08 | 0.07 |
| | Blackout+ScrambleEGO+LaneDeletion | 0.16 | 0.14 | 0.08 | 0.07 |
| | LaneDeletion | 0.20 | 0.17 | 0.11 | 0.05 |
| | RevertEGO | 0.13 | 0.10 | 0.13 | 0.06 |
| | ScrambleEGO | 0.21 | 0.19 | 0.13 | 0.06 |
| | ScrambleEGO+LaneDeletion | 0.23 | 0.21 | 0.15 | 0.05 |
| Dropout + LF | Original | 0.24 | 0.24 | 0.18 | 0.05 |
| | Blackout | 0.13 | 0.12 | 0.08 | 0.04 |
| | Blackout+ScrambleEGO+LaneDeletion | 0.18 | 0.18 | 0.10 | 0.04 |
| | LaneDeletion | 0.28 | 0.27 | 0.20 | 0.06 |
| | RevertEGO | 0.21 | 0.21 | 0.15 | 0.09 |
| | ScrambleEGO | 0.23 | 0.22 | 0.21 | 0.03 |
| | ScrambleEGO+LaneDeletion | 0.30 | 0.29 | 0.25 | 0.04 |
| PGP | Original | 0.17 | 0.17 | - | - |
| | Blackout | 0.09 | 0.09 | - | - |
| | Blackout+ScrambleEGO+LaneDeletion | 0.12 | 0.12 | - | - |
| | LaneDeletion | 0.14 | 0.14 | - | - |
| | RevertEGO | 0.15 | 0.15 | - | - |
| | ScrambleEGO | 0.14 | 0.14 | - | - |
| | ScrambleEGO+LaneDeletion | 0.13 | 0.13 | - | - |
| LP | Original | 0.16 | 0.16 | - | - |
| | Blackout | 0.15 | 0.15 | - | - |
| | Blackout+ScrambleEGO+LaneDeletion | 0.00 | 0.00 | - | - |
| | LaneDeletion | 0.06 | 0.06 | - | - |
| | RevertEGO | 0.15 | 0.15 | - | - |
| | ScrambleEGO | 0.16 | 0.16 | - | - |
| | ScrambleEGO+LaneDeletion | 0.13 | 0.13 | - | - |
| LF | Original | 0.22 | 0.22 | - | - |
| | Blackout | 0.04 | 0.04 | - | - |
| | Blackout+ScrambleEGO+LaneDeletion | 0.08 | 0.08 | - | - |
| | LaneDeletion | 0.23 | 0.23 | - | - |
| | RevertEGO | 0.20 | 0.20 | - | - |
| | ScrambleEGO | 0.25 | 0.25 | - | - |
| | ScrambleEGO+LaneDeletion | 0.24 | 0.24 | - | - |

Table 6: Pearson correlation coefficient between the different types of uncertainties (total, aleatoric, and epistemic) estimated by our approach and MinADE$_5$ across all out-of-distribution and the original dataset and all ensembles as well as single models.

| Model(s) | DS | Ours | | | Baseline |
|---|---|---|---|---|---|
| | | $\rho_{total}$ | $\rho_{aleatoric}$ | $\rho_{epistemic}$ | |
| $1 \times \{LP, LF, PGP\}$ | Original | 0.38 | 0.36 | 0.28 | 0.06 |
| | Blackout | 0.26 | 0.24 | 0.17 | 0.08 |
| | Blackout+ScrambleEGO+LaneDeletion | 0.24 | 0.21 | 0.16 | 0.13 |
| | LaneDeletion | 0.36 | 0.31 | 0.26 | 0.05 |
| | RevertEGO | 0.28 | 0.22 | 0.32 | 0.14 |
| | ScrambleEGO | 0.37 | 0.30 | 0.41 | 0.03 |
| | ScrambleEGO+LaneDeletion | 0.38 | 0.31 | 0.41 | 0.07 |
| $3 \times PGP$ | Original | 0.35 | 0.34 | 0.23 | 0.14 |
| | Blackout | 0.18 | 0.17 | 0.12 | 0.06 |
| | Blackout+ScrambleEGO+LaneDeletion | 0.25 | 0.20 | 0.24 | 0.18 |
| | LaneDeletion | 0.33 | 0.25 | 0.33 | 0.25 |
| | RevertEGO | 0.32 | 0.27 | 0.34 | 0.13 |
| | ScrambleEGO | 0.33 | 0.29 | 0.26 | 0.09 |
| | ScrambleEGO+LaneDeletion | 0.34 | 0.27 | 0.28 | 0.18 |
| $3 \times LP$ | Original | 0.27 | 0.19 | 0.28 | 0.11 |
| | Blackout | 0.17 | 0.14 | 0.12 | 0.11 |
| | Blackout+ScrambleEGO+LaneDeletion | 0.14 | 0.10 | 0.14 | 0.12 |
| | LaneDeletion | 0.21 | 0.12 | 0.31 | 0.14 |
| | RevertEGO | 0.10 | 0.05 | 0.24 | 0.03 |
| | ScrambleEGO | 0.37 | 0.31 | 0.33 | 0.04 |
| | ScrambleEGO+LaneDeletion | 0.33 | 0.26 | 0.34 | 0.03 |
| $3 \times LF$ | Original | 0.39 | 0.38 | 0.25 | 0.10 |
| | Blackout | 0.17 | 0.13 | 0.19 | 0.13 |
| | Blackout+ScrambleEGO+LaneDeletion | 0.25 | 0.22 | 0.22 | 0.09 |
| | LaneDeletion | 0.44 | 0.42 | 0.28 | 0.17 |
| | RevertEGO | 0.41 | 0.42 | 0.21 | 0.14 |
| | ScrambleEGO | 0.37 | 0.33 | 0.38 | 0.04 |
| | ScrambleEGO+LaneDeletion | 0.43 | 0.39 | 0.38 | 0.06 |
| Dropout + PGP | Original | 0.31 | 0.31 | 0.21 | 0.04 |
| | Blackout | 0.30 | 0.29 | 0.21 | 0.18 |
| | Blackout+ScrambleEGO+LaneDeletion | 0.22 | 0.18 | 0.23 | 0.17 |
| | LaneDeletion | 0.25 | 0.21 | 0.25 | 0.20 |
| | RevertEGO | 0.28 | 0.29 | 0.15 | 0.16 |
| | ScrambleEGO | 0.30 | 0.29 | 0.22 | 0.07 |
| | ScrambleEGO+LaneDeletion | 0.29 | 0.24 | 0.29 | 0.16 |
| Dropout + LP | Original | 0.21 | 0.15 | 0.23 | 0.17 |
| | Blackout | 0.21 | 0.15 | 0.20 | 0.14 |
| | Blackout+ScrambleEGO+LaneDeletion | 0.30 | 0.26 | 0.19 | 0.13 |
| | LaneDeletion | 0.33 | 0.28 | 0.21 | 0.09 |
| | RevertEGO | 0.12 | 0.08 | 0.20 | 0.14 |
| | ScrambleEGO | 0.31 | 0.26 | 0.25 | 0.10 |
| | ScrambleEGO+LaneDeletion | 0.32 | 0.28 | 0.23 | 0.10 |
| Dropout + LF | Original | 0.37 | 0.36 | 0.28 | 0.17 |
| | Blackout | 0.28 | 0.27 | 0.22 | 0.10 |
| | Blackout+ScrambleEGO+LaneDeletion | 0.35 | 0.34 | 0.23 | 0.10 |
| | LaneDeletion | 0.42 | 0.41 | 0.30 | 0.14 |
| | RevertEGO | 0.40 | 0.38 | 0.31 | 0.17 |
| | ScrambleEGO | 0.28 | 0.24 | 0.38 | 0.08 |
| | ScrambleEGO+LaneDeletion | 0.34 | 0.30 | 0.39 | 0.08 |
| PGP | Original | 0.27 | 0.27 | - | - |
| | Blackout | 0.10 | 0.10 | - | - |
| | Blackout+ScrambleEGO+LaneDeletion | 0.12 | 0.12 | - | - |
| | LaneDeletion | 0.17 | 0.17 | - | - |
| | RevertEGO | 0.20 | 0.20 | - | - |
| | ScrambleEGO | 0.27 | 0.27 | - | - |
| | ScrambleEGO+LaneDeletion | 0.16 | 0.16 | - | - |
| LP | Original | 0.15 | 0.15 | - | - |
| | Blackout | 0.19 | 0.19 | - | - |
| | Blackout+ScrambleEGO+LaneDeletion | 0.07 | 0.07 | - | - |
| | LaneDeletion | -0.01 | -0.01 | - | - |
| | RevertEGO | 0.07 | 0.07 | - | - |
| | ScrambleEGO | 0.11 | 0.11 | - | - |
| | ScrambleEGO+LaneDeletion | 0.10 | 0.10 | - | - |
| LF | Original | 0.26 | 0.26 | - | - |
| | Blackout | 0.07 | 0.07 | - | - |
| | Blackout+ScrambleEGO+LaneDeletion | 0.15 | 0.15 | - | - |
| | LaneDeletion | 0.29 | 0.29 | - | - |
| | RevertEGO | 0.27 | 0.27 | - | - |
| | ScrambleEGO | 0.27 | 0.27 | - | - |
| | ScrambleEGO+LaneDeletion | 0.31 | 0.31 | - | - |

Table 7: Pearson correlation coefficient between the different types of uncertainties (total, aleatoric, and epistemic) estimated by our approach and MinADE$_{10}$ across all out-of-distribution and the original dataset and all ensembles as well as single models.

| Model(s) | DS | Ours | | | Baseline |
|---|---|---|---|---|---|
| | | $\rho_{total}$ | $\rho_{aleatoric}$ | $\rho_{epistemic}$ | |
| $1 \times \{LP, LF, PGP\}$ | Original | 0.38 | 0.36 | 0.28 | 0.08 |
| | Blackout | 0.23 | 0.21 | 0.15 | 0.06 |
| | Blackout+ScrambleEGO+LaneDeletion | 0.23 | 0.20 | 0.15 | 0.13 |
| | LaneDeletion | 0.37 | 0.32 | 0.24 | 0.07 |
| | RevertEGO | 0.25 | 0.21 | 0.27 | 0.17 |
| | ScrambleEGO | 0.42 | 0.36 | 0.39 | 0.05 |
| | ScrambleEGO+LaneDeletion | 0.39 | 0.33 | 0.39 | 0.07 |
| $3 \times PGP$ | Original | 0.33 | 0.31 | 0.24 | 0.17 |
| | Blackout | 0.13 | 0.12 | 0.08 | 0.05 |
| | Blackout+ScrambleEGO+LaneDeletion | 0.23 | 0.17 | 0.25 | 0.14 |
| | LaneDeletion | 0.33 | 0.25 | 0.35 | 0.25 |
| | RevertEGO | 0.25 | 0.20 | 0.31 | 0.12 |
| | ScrambleEGO | 0.33 | 0.29 | 0.25 | 0.07 |
| | ScrambleEGO+LaneDeletion | 0.33 | 0.26 | 0.30 | 0.22 |
| $3 \times LP$ | Original | 0.25 | 0.16 | 0.30 | 0.12 |
| | Blackout | 0.14 | 0.12 | 0.08 | 0.14 |
| | Blackout+ScrambleEGO+LaneDeletion | 0.10 | 0.07 | 0.11 | 0.13 |
| | LaneDeletion | 0.18 | 0.09 | 0.32 | 0.16 |
| | RevertEGO | 0.10 | 0.07 | 0.19 | 0.00 |
| | ScrambleEGO | 0.35 | 0.28 | 0.31 | 0.04 |
| | ScrambleEGO+LaneDeletion | 0.32 | 0.26 | 0.31 | 0.04 |
| $3 \times LF$ | Original | 0.39 | 0.38 | 0.24 | 0.11 |
| | Blackout | 0.13 | 0.09 | 0.17 | 0.09 |
| | Blackout+ScrambleEGO+LaneDeletion | 0.22 | 0.19 | 0.20 | 0.10 |
| | LaneDeletion | 0.44 | 0.43 | 0.28 | 0.15 |
| | RevertEGO | 0.40 | 0.40 | 0.19 | 0.16 |
| | ScrambleEGO | 0.37 | 0.32 | 0.39 | 0.05 |
| | ScrambleEGO+LaneDeletion | 0.44 | 0.40 | 0.38 | 0.07 |
| Dropout + PGP | Original | 0.30 | 0.30 | 0.20 | 0.06 |
| | Blackout | 0.30 | 0.29 | 0.21 | 0.18 |
| | Blackout+ScrambleEGO+LaneDeletion | 0.20 | 0.16 | 0.23 | 0.17 |
| | LaneDeletion | 0.23 | 0.18 | 0.27 | 0.05 |
| | RevertEGO | 0.27 | 0.27 | 0.14 | 0.13 |
| | ScrambleEGO | 0.32 | 0.30 | 0.23 | 0.07 |
| | ScrambleEGO+LaneDeletion | 0.28 | 0.23 | 0.29 | 0.11 |
| Dropout + LP | Original | 0.19 | 0.12 | 0.25 | 0.19 |
| | Blackout | 0.20 | 0.14 | 0.22 | 0.17 |
| | Blackout+ScrambleEGO+LaneDeletion | 0.31 | 0.27 | 0.20 | 0.15 |
| | LaneDeletion | 0.32 | 0.26 | 0.23 | 0.11 |
| | RevertEGO | 0.13 | 0.09 | 0.19 | 0.17 |
| | ScrambleEGO | 0.26 | 0.22 | 0.24 | 0.13 |
| | ScrambleEGO+LaneDeletion | 0.28 | 0.24 | 0.23 | 0.12 |
| Dropout + LF | Original | 0.37 | 0.36 | 0.29 | 0.21 |
| | Blackout | 0.31 | 0.30 | 0.25 | 0.14 |
| | Blackout+ScrambleEGO+LaneDeletion | 0.37 | 0.36 | 0.25 | 0.13 |
| | LaneDeletion | 0.43 | 0.41 | 0.31 | 0.16 |
| | RevertEGO | 0.41 | 0.39 | 0.32 | 0.19 |
| | ScrambleEGO | 0.24 | 0.18 | 0.41 | 0.08 |
| | ScrambleEGO+LaneDeletion | 0.29 | 0.24 | 0.41 | 0.06 |
| PGP | Original | 0.29 | 0.29 | - | - |
| | Blackout | 0.07 | 0.07 | - | - |
| | Blackout+ScrambleEGO+LaneDeletion | 0.13 | 0.13 | - | - |
| | LaneDeletion | 0.16 | 0.16 | - | - |
| | RevertEGO | 0.20 | 0.20 | - | - |
| | ScrambleEGO | 0.25 | 0.25 | - | - |
| | ScrambleEGO+LaneDeletion | 0.15 | 0.15 | - | - |
| LP | Original | 0.12 | 0.12 | - | - |
| | Blackout | 0.16 | 0.16 | - | - |
| | Blackout+ScrambleEGO+LaneDeletion | 0.01 | 0.01 | - | - |
| | LaneDeletion | -0.07 | -0.07 | - | - |
| | RevertEGO | 0.10 | 0.10 | - | - |
| | ScrambleEGO | 0.12 | 0.12 | - | - |
| | ScrambleEGO+LaneDeletion | 0.10 | 0.10 | - | - |
| LF | Original | 0.39 | 0.39 | - | - |
| | Blackout | 0.09 | 0.09 | - | - |
| | Blackout+ScrambleEGO+LaneDeletion | 0.18 | 0.18 | - | - |
| | LaneDeletion | 0.42 | 0.42 | - | - |
| | RevertEGO | 0.40 | 0.40 | - | - |
| | ScrambleEGO | 0.30 | 0.30 | - | - |
| | ScrambleEGO+LaneDeletion | 0.35 | 0.35 | - | - |

## I    ADDITIONAL ERROR PLOTS FOR OOD SCENARIOS

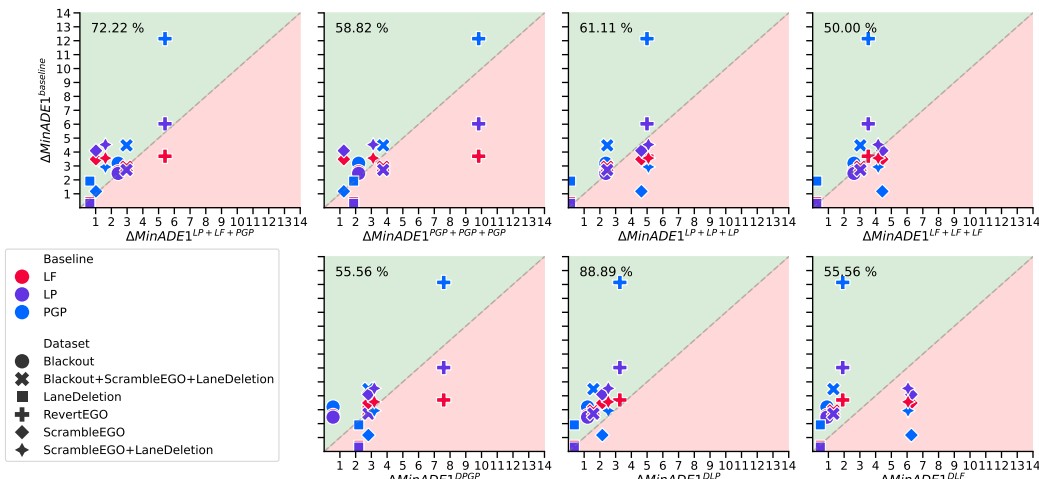

Figure 17: Differences ($\Delta$) in MinADE$_1$ between the original dataset and the corresponding out-of-distribution dataset for baseline models (y-axis) and ensembles (x-axis) over the validation set. Different colors correspond to various baseline models, while different markers denote distinct datasets.

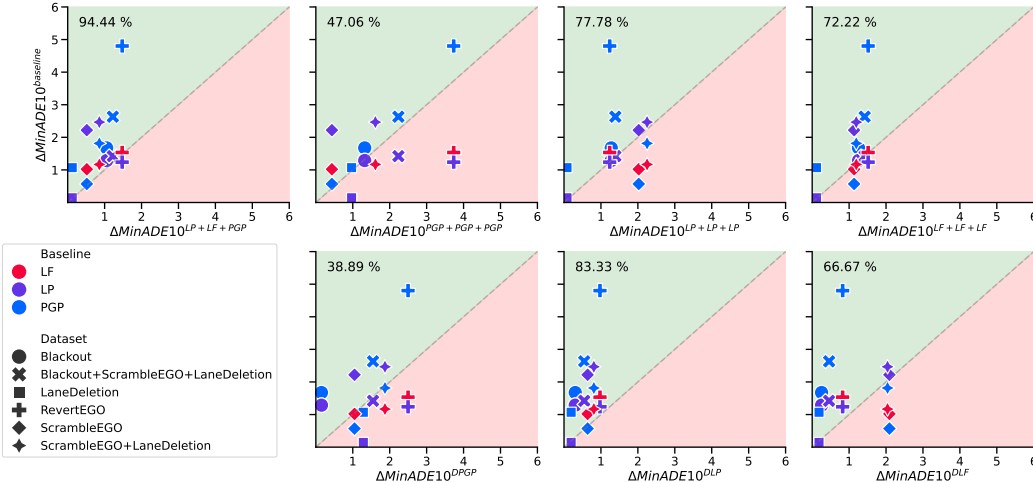

Figure 18: Differences ($\Delta$) in MinADE$_{10}$ between the original dataset and the corresponding out-of-distribution dataset for baseline models (y-axis) and ensembles (x-axis) over the validation set. Different colors correspond to various baseline models, while different markers denote distinct datasets.

## J    VALUES OF ERROR CHANGES IN OOD SCENARIOS

Table 8: Differences ($\Delta$) in MinADE$_k$ and MinFDE$_k$ between the corresponding out-of-distribution dataset and original dataset for PGP, Dropout + PGP, and $3 \times$ PGP.

| Dataset | Model(s) | $\Delta$MinADE($\downarrow$) | | | $\Delta$MinFDE($\downarrow$) | | |
|---|---|---|---|---|---|---|---|
| | | $k = 1$ | $k = 5$ | $k = 10$ | $k = 1$ | $k = 5$ | $k = 10$ |
| Blackout | PGP | 3.20 | 2.20 | 1.68 | 5.62 | 3.66 | 2.56 |
| | Dropout + PGP | **0.57** | **0.22** | **0.15** | **1.06** | **0.31** | **0.17** |
| | $3 \times$ PGP | 2.19 | 1.49 | 1.32 | 3.39 | 2.00 | 1.70 |
| Blackout | PGP | 4.48 | 3.35 | 2.63 | 8.58 | 6.53 | 5.06 |
| + ScrambleEGO | Dropout + PGP | **2.82** | **1.78** | **1.55** | **6.17** | **4.18** | **3.71** |
| + LaneDeletion | $3 \times$ PGP | 3.75 | 2.50 | 2.24 | 7.00 | 4.53 | 4.03 |
| LaneDeletion | PGP | 1.91 | 1.41 | 1.07 | 4.48 | 3.61 | 2.83 |
| | Dropout + PGP | 2.20 | 1.46 | 1.29 | 5.07 | 3.65 | 3.32 |
| | $3 \times$ PGP | **1.87** | **1.14** | **0.97** | **4.36** | **2.87** | **2.52** |
| RevertEGO | PGP | 12.15 | 6.95 | 4.80 | 20.47 | 11.19 | 7.34 |
| | Dropout + PGP | **7.60** | **3.15** | **2.50** | **12.54** | **4.50** | **3.42** |
| | $3 \times$ PGP | 9.82 | 4.52 | 3.73 | 15.91 | 6.55 | 5.22 |
| ScrambleEGO | PGP | **1.18** | 0.76 | 0.57 | **1.69** | 0.91 | 0.61 |
| | Dropout + PGP | 2.81 | 1.31 | 1.05 | 3.91 | 1.42 | 0.95 |
| | $3 \times$ PGP | 1.25 | **0.55** | **0.43** | 1.76 | **0.65** | **0.50** |
| ScrambleEGO | PGP | **2.93** | 2.28 | 1.81 | **5.78** | 4.63 | 3.68 |
| + LaneDeletion | Dropout + PGP | 3.19 | 2.08 | 1.87 | 5.82 | **3.86** | 3.43 |
| | $3 \times$ PGP | 3.11 | **1.86** | **1.61** | 6.04 | **3.42** | 3.89 |

Table 9: Differences ($\Delta$) in MinADE$_k$ and MinFDE$_k$ between the corresponding out-of-distribution dataset and original dataset for LF, Dropout + LF, and $3 \times$ LF.

| Dataset | Model(s) | $\Delta$MinADE($\downarrow$) | | | $\Delta$MinFDE($\downarrow$) | | |
|---|---|---|---|---|---|---|---|
| | | $k = 1$ | $k = 5$ | $k = 10$ | $k = 1$ | $k = 5$ | $k = 10$ |
| Blackout | LF | 2.58 | 2.15 | 1.29 | 3.76 | 2.85 | 1.12 |
| | Dropout + LF | **0.91** | **0.25** | 0.38 | **1.87** | **0.44** | 0.74 |
| | $3 \times$ LF | 2.63 | 1.40 | 1.26 | 3.96 | 1.52 | 1.15 |
| Blackout | LF | 2.94 | 1.42 | 2.38 | 4.63 | 3.46 | 1.57 |
| + ScrambleEGO | Dropout + LF | **1.32** | **0.46** | **0.60** | **2.78** | **0.98** | **1.29** |
| + LaneDeletion | $3 \times$ LF | 3.03 | 1.42 | 1.59 | 4.82 | 1.61 | 2.02 |
| LaneDeletion | LF | 0.45 | 0.13 | 0.18 | 1.13 | 0.37 | 0.51 |
| | Dropout + LF | 0.49 | 0.19 | 0.21 | 1.23 | 0.51 | 0.57 |
| | $3 \times$ LF | **0.26** | **0.12** | **0.14** | **0.68** | **0.32** | **0.38** |
| RevertEGO | LF | 3.70 | 1.54 | 2.87 | 5.59 | 1.87 | 4.30 |
| | Dropout + LF | **1.91** | **0.83** | **0.95** | **2.86** | **1.07** | **1.32** |
| | $3 \times$ LF | 3.53 | 1.52 | 1.72 | 5.28 | 1.95 | 2.30 |
| ScrambleEGO | LF | **3.48** | **1.02** | 2.54 | **5.53** | **0.91** | 3.62 |
| | Dropout + LF | 6.27 | 2.09 | 2.59 | 7.83 | 1.94 | 2.72 |
| | $3 \times$ LF | 4.43 | 1.14 | **1.42** | 6.73 | 0.99 | **1.46** |
| ScrambleEGO | LF | **3.56** | **1.17** | 2.34 | **6.00** | 1.27 | 3.36 |
| + LaneDeletion | Dropout + LF | 6.06 | 2.04 | 2.49 | 7.74 | 2.12 | 2.80 |
| | $3 \times$ LF | 4.16 | 1.19 | **1.41** | 6.22 | **1.18** | **1.56** |

Table 10: Differences ($\Delta$) in MinADE$_k$ and MinFDE$_k$ between the corresponding out-of-distribution dataset and original dataset for LP, Dropout + LP, and $3 \times$ LP.

| Dataset | Model(s) | $\Delta$MinADE($\downarrow$) | | | $\Delta$MinFDE($\downarrow$) | | |
|---|---|---|---|---|---|---|---|
| | | $k=1$ | $k=5$ | $k=10$ | $k=1$ | $k=5$ | $k=10$ |
| Blackout | LP | 2.46 | 1.76 | 1.29 | 4.29 | 2.68 | 1.69 |
| | Dropout + LP | **1.22** | **0.39** | **0.30** | **2.41** | **0.67** | **0.48** |
| | $3 \times$ LP | 2.38 | 1.38 | 1.28 | 4.03 | 1.92 | 1.71 |
| Blackout | LP | 2.72 | 1.88 | 1.42 | 4.88 | 3.02 | 2.04 |
| + ScrambleEGO | Dropout + LP | **1.59** | **0.66** | **0.55** | **3.25** | **1.33** | **1.09** |
| + LaneDeletion | $3 \times$ LP | 2.47 | 1.51 | 1.39 | 4.27 | 2.23 | 1.96 |
| LaneDeletion | LP | 0.31 | 0.17 | 0.14 | 0.77 | 0.42 | 0.37 |
| | Dropout + LP | 0.38 | 0.21 | 0.19 | 0.86 | 0.53 | 0.49 |
| | $3 \times$ LP | **0.14** | **0.10** | **0.08** | **0.35** | **0.25** | **0.21** |
| RevertEGO | LP | 6.03 | 2.01 | 1.24 | 10.97 | 3.36 | 1.74 |
| | Dropout + LP | **3.27** | **1.14** | **0.97** | **5.54** | **1.87** | **1.49** |
| | $3 \times$ LP | 4.99 | 1.61 | 1.24 | 8.61 | 2.65 | 1.89 |
| ScrambleEGO | LP | 4.10 | 2.41 | 2.22 | 7.10 | 2.73 | 2.34 |
| | Dropout + LP | **2.16** | **0.80** | **0.64** | **3.30** | **0.94** | **0.61** |
| | $3 \times$ LP | 4.64 | 2.36 | 2.02 | 6.94 | 2.81 | 2.25 |
| ScrambleEGO | LP | 4.53 | 2.71 | 2.47 | 7.91 | 3.42 | 2.91 |
| + LaneDeletion | Dropout + LP | **2.54** | **1.03** | **0.81** | **4.19** | **1.51** | **1.05** |
| | $3 \times$ LP | 5.08 | 2.64 | 2.25 | 7.88 | 3.46 | 2.76 |

Table 11: Differences ($\Delta$) in MinADE$_k$ and MinFDE$_k$ between the corresponding out-of-distribution dataset and original dataset for various models, including LP, LF, and PGP, as well as their combination $1 \times \{$LP, LF, PGP$\}$.

| Dataset | Model(s) | $\Delta$MinADE | | | $\Delta$MinFDE | | |
|---|---|---|---|---|---|---|---|
| | | $k=1$ | $k=5$ | $k=10$ | $k=1$ | $k=5$ | $k=10$ |
| Blackout | PGP | 3.20 | 2.20 | 1.68 | 5.62 | 3.66 | 2.56 |
| | LF | 2.58 | 2.15 | 1.29 | **3.76** | 2.85 | **1.12** |
| | LP | 2.46 | 1.76 | 1.29 | 4.29 | 2.68 | 1.69 |
| | $1 \times \{$LP, LF, PGP$\}$ | **2.43** | **1.18** | **1.06** | 3.95 | **1.58** | 1.35 |
| Blackout | PGP | 4.48 | 3.35 | 2.63 | 8.58 | 6.53 | 5.06 |
| + ScrambleEGO | LF | 2.94 | 2.38 | 1.42 | **4.63** | 3.46 | **1.57** |
| + LaneDeletion | LP | **2.72** | 1.88 | 1.42 | 4.88 | 3.02 | 2.04 |
| | $1 \times \{$LP, LF, PGP$\}$ | 2.96 | **1.38** | **1.22** | 5.18 | **2.05** | 1.75 |
| LaneDeletion | PGP | 1.91 | 1.41 | 1.07 | 4.48 | 3.61 | 2.83 |
| | LF | 0.45 | 0.18 | 0.13 | 1.13 | 0.51 | 0.37 |
| | LP | **0.31** | **0.17** | 0.14 | **0.77** | **0.42** | 0.37 |
| | $1 \times \{$LP, LF, PGP$\}$ | 0.63 | **0.16** | **0.12** | 1.50 | 0.44 | **0.34** |
| RevertEGO | PGP | 12.15 | 6.95 | 4.80 | 20.47 | 11.19 | 7.34 |
| | LF | **3.70** | 2.87 | 1.54 | **5.59** | 4.30 | 1.87 |
| | LP | 6.03 | 2.01 | **1.24** | 10.97 | 3.36 | **1.74** |
| | $1 \times \{$LP, LF, PGP$\}$ | 5.41 | **1.88** | 1.48 | 8.81 | **2.95** | 2.14 |
| ScrambleEGO | PGP | 1.18 | 0.76 | 0.57 | 1.69 | **0.91** | 0.61 |
| | LF | 3.48 | 2.54 | 1.02 | 5.53 | 3.62 | 0.91 |
| | LP | 4.10 | 2.41 | 2.22 | 7.10 | 2.73 | 2.34 |
| | $1 \times \{$LP, LF, PGP$\}$ | **1.02** | **0.75** | **0.52** | **1.57** | 1.00 | **0.58** |
| ScrambleEGO | PGP | 2.93 | 2.28 | 1.81 | 5.78 | 4.63 | 3.68 |
| + LaneDeletion | LF | 3.56 | 2.34 | 1.17 | 6.00 | 3.36 | 1.27 |
| | LP | 4.53 | 2.71 | 2.47 | 7.91 | 3.42 | 2.91 |
| | $1 \times \{$LP, LF, PGP$\}$ | **1.62** | **1.14** | **0.86** | **2.93** | **1.76** | **1.17** |

