# OpenReview forum: "Entropy-Based Uncertainty Modeling for Trajectory Prediction in Autonomous Driving"
_ICLR.cc/2025/Conference — Submitted to ICLR 2025_

### Official Review · Reviewer_Paxb · 2024-11-02

**Soundness:** 2
**Presentation:** 2
**Contribution:** 2
**Rating:** 5
**Confidence:** 4

**Summary:**

This paper presents an entropy-based approach for modeling uncertainty in trajectory prediction for autonomous driving, leveraging an information-theoretic framework to decompose total uncertainty into aleatoric and epistemic components. It conducts extensive experiments on the nuScenes dataset, analyzing the relationship between uncertainty and prediction error, and evaluates model robustness in both in- and out-of-distribution scenarios, contributing a novel method for assessing and improving the reliability of trajectory prediction models in autonomous vehicles.

**Strengths:**

This paper presents a pioneering approach to uncertainty quantification in autonomous driving trajectory prediction by introducing an information-theoretic framework that decomposes total uncertainty into aleatoric and epistemic components. This method, supported by extensive experiments on the nuScenes dataset, not only enhances the safety and reliability of autonomous vehicles but also advances the field by providing a more nuanced understanding of prediction uncertainties, which is crucial for making informed decisions in complex traffic scenarios.

**Weaknesses:**

This paper on entropy-based uncertainty modeling for autonomous driving trajectory prediction presents a significant advancement with its information-theoretic approach to quantify and decompose uncertainty, but could be further strengthened by expanding experimental validation to more diverse datasets, exploring additional model architectures, addressing computational efficiency, deepening the theoretical foundation, and demonstrating practical integration with planning systems to enhance its applicability and impact in the field.

**Questions:**

This paper primarily utilizes the nuScenes dataset for experiments. Could the authors comment on the generalizability of their approach when applied to other datasets that may have different characteristics or geographical locations?
This paper mentions the increased computational burden of the proposed method. What specific strategies are being considered to optimize computational efficiency, especially for real-time applications?
This paper could benefit from experimenting with a broader range of neural network architectures. Have the authors considered or tested other architectures, and if so, what were the findings?
This paper could be strengthened by a more rigorous theoretical analysis of the uncertainty decomposition process. Are there any ongoing efforts to develop a more formal theoretical framework?
This paper suggests the potential integration of the uncertainty model with planning systems. Could the authors elaborate on any initial experiments or simulations that demonstrate this integration?
This paper mentions the application of uncertainty estimates in risk-sensitive reinforcement learning. Could the authors provide more details on how these estimates could inform decision-making processes under risk?

---

> ### Author Response · Authors · 2024-11-24
>
> We thank the reviewer for the constructive comments on our paper. Regarding the concerns of the reviewer Paxb, we provide the following responses.
>
> Q1: As our approach does not depend on the dataset it is expected to also work for other datasets.
>
> Q2: The computation burden is in fact a downside of this approach, possible extensions to tackle this issue are provided in the discussion (Section 2.3).
>
> Q3: The paper could be expanded by evaluating additional architectures; however, testing an infinite number of architectures is impossible. As a result, we had to limit our evaluation to a selected set of architectures. Nonetheless, the approach remains theoretically grounded.
>
> Q4: There are no initial experiments for integrating the method into a planner, as this falls outside the scope of this paper. This could be explored as future work.

---

### Official Review · Reviewer_7w8N · 2024-11-03

**Soundness:** 2
**Presentation:** 3
**Contribution:** 1
**Rating:** 5
**Confidence:** 4

**Summary:**

The paper focuses on quantifying and decomposing uncertainty into epistemic and aleatoric components for trajectory prediction in autonomous driving. It measures aleatoric uncertainty using conditional entropy and epistemic uncertainty through mutual information. The authors employ a Monte Carlo approximation to estimate these uncertainties and evaluate their approach on the nuScenes dataset, comparing it against a baseline method that quantifies only epistemic uncertainty.

**Strengths:**

The authors show that by separating aleatoric uncertainty, they can achieve a more precise estimation of epistemic uncertainty, which generalizes well to motion forecasting. Validated on the nuScenes dataset, the approach quantifies uncertainty across both in-distribution and out-of-distribution scenarios. In out-of-distribution scenarios, the authors show that mixed ensembles, which combine various model architectures, enhance reliability.

The proposed approach is evaluated across various scenarios, with additional results for different model variations provided in the appendix.

**Weaknesses:**

* **W1:** The authors state that they believe they are the first to thoroughly investigate quantification of uncertainties for trajectory prediction models. However, after a quick research, I found that M. Itkina, "Interpretable Self-Aware Neural Networks for Robust Trajectory Prediction", CoRL, 2022 also break down prediction uncertainties in aleotoric and epistemic components. Could you compare your approach to Itkina et al.'s method, highlighting key differences or improvements?

* **W2:** The paper compares the proposed method against only one baseline: Robust Imitative Planning (RIP) by Filos et al. "Can Autonomous Vehicles Identify, Recover From, and Adapt to Distribution Shifts?", 2020. RIP is a heuristic-based approach, yet the paper lacks details on how RIP was used. Given that RIP was originally developed for a different dataset, its default parameters may not be optimally suited for the test cases used here.

  * **W2a:** Together with **M4** below, comparing your proposed method on the CARNOVEL benchmark could highlight its contribution.

  * **W2b:** Please specify how RIP was adapted or tuned for the dataset used in this study.

* **W3:** The underlying prediction work of this approach, Model-Based Risk Minimization (MBRM), is a work currently under review for IEEE ICRA 2025. MBRM uses the same ensembles (deep ensembles and dropout ensembles) used in this work. While a clear differentiation in the paper body is lacking, the main differences between this work and MBRM appear to be the use of Monte Carlo-based uncertainty estimation and the inclusion of OOD scenarios.

* **W4:** The analysis in Section 3.4, 'Detecting OOD Scenarios,' lacks a clear takeaway. While it shows that the mixed deep ensemble of LAformer, PGP, and LaPred assigns higher epistemic uncertainty to out-of-distribution datasets, a comparison with common OOD detection baselines (e.g. Mahalanobis distance, ensemble disagreement, or AUROC) would clarify the effectiveness of the proposed approach.

The novelty of this work appears incremental, primarily applying existing ideas to motion prediction rather than introducing new concepts or hypotheses. Additionally, the paper does not focus on representation learning, and its quantitative comparisons with baselines lack depth.

**Questions:**

**Questions:**
* **Q1:** The main baseline you compare your approach against is Robust Imitative Planning (RIP). Because that approach was developed for another dataset, can you clarify how exactly you used that approach in your test cases (as mentioned in **W2b**)?
* **Q2:** How reproducible is your approach? Did you conduct any hyperparameter analysis, or do you plan to open-source your implementation?
* **Q3:** You presented results for minADE, and I would expect similar trends for minFDE. If you analyzed the correlation and results for minFDE, including these or summarizing them in a single sentence could provide valuable insight for some readers.

**Improvement suggestions:**
* **S1:** Make a clear distinction throughout the text between the contributions of this work and elements borrowed from "Motion Forecasting via Model-Based Risk Minimization". One example is line 237.
* **S2:** Lines 245-249 disrupt the overview flow and should be revised for clarity.
* **S3:** In abstract you state: "_To ensure safety, planners must rely on reliable uncertainty information about the predicted future behavior of surrounding agents, yet this aspect has received limited attention. This paper addresses the so-far neglected problem of uncertainty modeling in trajectory prediction._" However, there are many approaches that address perception and prediction uncertainties in the environment. There are also dozens of approaches that predict the uncertainty, though often using a different method than yours. Therefore, these sentences must be revised for clarity.

**Minor suggestions:**
* **M1:** Consider adding a clear statement on whether the uncertainty quantification impacts prediction accuracy.
* **M2:** You used MBRM to sample trajectories from an ensemble of prediction models. Does your uncertainty quantification approach impose any specific requirements on the prediction model used? For instance, is fitting a Gaussian Mixture Model (GMM) necessary? Additionally, can this approach be used with arbitrary prediction methods?
* **M3:** Adding N. Kose, "Reliable Multimodal Trajectory Prediction via Error Aligned Uncertainty Optimization," ECCV, 2022, to related work might be helpful.
* **M4:** One last point: application-focused papers are typically expected to demonstrate a quantifiable, measurable advantage over baselines. For example, as in RIP, applying a planning benchmark to show how improved epistemic uncertainty estimates enhance the robustness of baseline planning algorithms could be beneficial. While not strictly necessary, such an analysis would strengthen the paper by providing concrete evidence of practical benefits.

---

> ### Author Response · Authors · 2024-11-24
>
> We thank the reviewer for the constructive comments on our paper. Regarding the concerns of the reviewer 7w8N, we provide the following responses.
>
> Q1: We substituted each model in the ensemble with its corresponding Gaussian mixture model. Using these Gaussian mixture models, we calculate the log probability of the ground truth and compute the variance across these values. The formula itself remains unchanged; only the models and data have changed.
>
> Q2: We plan to open source our implementation. Besides that, no sensitive hyperparameters are involved and therefore the approach can be easily reproducible.
>
> Q3: We also evaluated the correlation for minFDE and observed similar results to minADE, as minADE is the more meaningful metric. Therefore, we chose to omit the minFDE results.

---

> ### Comment · Reviewer_7w8N · 2024-11-25
>
> Thank you for your clarifications.
>
> I was curious about the sensitivity to the number of samples (N). Your explanation is helpful and addresses much of my concern. I appreciate that you plan to open-source your implementation.
>
> Do you have any comments or revisions regarding "W1" in the weaknesses list?

---

> > ### Author Response · Authors · 2024-11-26
> >
> > Good point! We had considered that as well but haven’t tested the baseline with varying numbers of samples, as it would require large ensembles. However, this could reveal another potential weakness of the baseline.
> >
> > Regarding W1: Thank you for looking into that. You are correct that Itkina et al. also decompose uncertainty into epistemic and aleatoric. However, they employ a different approach (evidential deep learning) compared to our theoretically grounded method.

---

> > > ### Comment · Reviewer_7w8N · 2024-11-26
> > >
> > > Please add those remarks in the paper (both the rationale for choosing minADE over minFDE and a discussion of Itkina's use of evidential deep learning in the related work section) -- in case, for the camera ready version.
> > >
> > > It would also be helpful if you could demonstrate, at least for your approach, how the uncertainty estimation changes with varying numbers of samples.
> > >
> > > Could you please specify which RIP variant you used ("WCM", "MA", "BCM")?
> > >
> > > Does a comparison with OOD detection baselines such as AUROC make sense? If not, could you clarify why?

---

> > > > ### Author Response · Authors · 2024-11-27
> > > >
> > > > In our approach, the number of samples is a configurable parameter. We experimented with various numbers and selected the value beyond which the uncertainty estimation stabilized and showed no significant improvement.
> > > >
> > > > Since our focus was on prediction rather than a control task, we did not utilize WC, MA, or BCM. Instead, we employed their equation for uncertainty estimation, as described in Equation 3 of the RIP paper.
> > > >
> > > > Given that this is a regression setting, using AUROC would not be appropriate in this context.

---

> > > > > ### Comment · Reviewer_7w8N · 2024-12-03
> > > > >
> > > > > Thank you for your detailed explanations; they have improved my understanding and evaluation of your contributions. I encourage you to incorporate the remarks discussed so far into the revised version of your work. Additionally, including an analysis of how the number of samples used affects the uncertainty estimation would be interesting.
> > > > >
> > > > > I realized that my initial assessment was somewhat off. Therefore, during the rebuttal, I've increased my rating from 3 (reject) to 5 (marginally below the acceptance threshold). While your approach is theoretically more grounded than previous work and not tied to a specific model, this is not directly compared within the paper. I'd recommend a direct, scenario-based quantitative comparison, as this would better highlight your contributions.
> > > > >
> > > > > While I welcome a discussion on how the decomposed uncertainty components could be utilized within the driving stack, I do not consider it absolutely necessary for this paper. Demonstrating how uncertainty quantification directly impacts downstream performance -- such as planning or decision-making in autonomous driving -- could be left to future work.
> > > > >
> > > > > The proposed Monte Carlo-based uncertainty estimation requires a more in-depth analysis of its limitations. This should include performance across datasets or driving scenarios and an analysis of sensitivities, such as the effect of varying number of samples on the uncertainty estimation.

---

### Official Review · Reviewer_ibRw · 2024-11-04

**Soundness:** 2
**Presentation:** 3
**Contribution:** 2
**Rating:** 5
**Confidence:** 3

**Summary:**

This paper claims to propose a method to measure and decompose the uncertainty of trajectory prediction models, analyze the relationship between uncertainty and prediction error when the scenarios are either in-distribution or out-of-distribution, study how different configurations of ensemble compositions affect the uncertainty quantification and model robustness.

**Strengths:**

1. It is interesting to see the idea of quantifying and decomposing the uncertainty of trajectory prediction models.
2. The experiments results show that most of the findings through the proposed approach aligns with the existing research conclusions.

**Weaknesses:**

The biggest concern here, as what it is mentioned in the discussion, is the memory and computational burden. The inference time to sample from the ensembles, GPUs, memory usage used in the experiments are not reported. Even if we consider the distilling ensembles into one model, it also takes extra computations.

**Questions:**

1. In table 1, why is $3\times LP$'s total uncertainty has lower correlation with the prediction error than its individual components? Can you provide some insights for it?
2. is this proposed approach a means for evaluating model uncertainty? Any other real-world applications for this method?
3. As most of the findings are aligned with the known conclusions from the existing researches, is this approach simply for ensuring the conclusions are correct?

---

> ### Author Response · Authors · 2024-11-24
>
> We really appreciate the reviewer for the constructive comments and feedback on our paper.
>
> Q1: The 3xLP (deep ensemble) demonstrates the highest total correlation compared to both the 3xLP (dropout ensemble) and the 1xLP. This trend is consistent across other architectures and is attributed to deep ensembles producing the most diverse predictions. In contrast, models within dropout ensembles are more prone to converging to similar local minima, resulting in less diversity.
>
> Q2: This paper explores this uncertainty quantification approach within the context of autonomous driving. However, the approach is not confined to this domain and holds potential for application in other areas in future research.
>
> Q3: Uncertainty quantification remains relatively unexplored in autonomous driving. This paper introduces an approach for uncertainty quantification that has not been applied in this domain before. The primary goal is not merely to confirm existing conclusions but to demonstrate how effectively this method can quantify uncertainty in autonomous driving.

---

> > ### Comment · Reviewer_ibRw · 2024-12-03
> >
> > Thank you for your response and I decided to keep my score.

---

### Official Review · Reviewer_5JQ7 · 2024-11-04

**Soundness:** 2
**Presentation:** 3
**Contribution:** 2
**Rating:** 3
**Confidence:** 4

**Summary:**

This paper explores uncertainty modeling in trajectory prediction for autonomous driving, utilizing information-theoretic approaches to decompose uncertainty into aleatoric and epistemic components. Experiments are conducted on the nuScenes dataset to analyze correlations between prediction errors and various uncertainty types. Additionally, the paper examines model robustness in out-of-distribution (OOD) scenarios and investigates techniques for effectively detecting OOD scenarios.

**Strengths:**

1. The paper is well-presented, with a clear structure and highly informative images that effectively support the content.
2. The paper has comprehensive coverage of relevant prior work and contextual background.

**Weaknesses:**

1. The primary critique of this paper is that, despite its thorough analysis of uncertainty decomposition, the relationships between prediction errors and uncertainty components, and so on, it fails to address how these insights could be leveraged to reduce prediction errors. While uncertainty decomposition provides valuable interpretative insights, the autonomous driving community is more focused on actionable strategies for minimizing prediction error. Unfortunately, the paper does not propose any methods to achieve this.
2. In Section 3.4, the paper leverages uncertainty to identify out-of-distribution (OOD) scenarios, claiming that "it facilitates the collection of challenging cases for re-training and evaluation." However, in the autonomous driving community, practitioners typically focus on scenarios with the highest prediction error. The paper would benefit from justifying why using uncertainty to identify OOD scenarios offers advantages for training and evaluation compared to simply targeting high-error scenarios.
3. The paper’s analysis is limited to a selection of trajectory prediction models and their ensemble versions, making it difficult to generalize the findings to other model architectures.

**Questions:**

1. The paper would be strengthened by including examples demonstrating how uncertainty analysis can directly contribute to reducing prediction error.
2. The paper could be strengthened by providing justification for how its findings on certain models might generalize to other model architectures.

---

> ### Author Response · Authors · 2024-11-24
>
> We thank the reviewer for the constructive comments. Regarding the concerns of the reviewer 5JQ7, we provide the following responses.
>
> Q1: The paper demonstrates that epistemic uncertainty is higher in out-of-distribution (OOD) scenarios, helping detect when the model is less confident in its predictions (Section 3.4). A planner could leverage these high-uncertainty instances to avoid risky maneuvers. Possible future works based on this approach are mentioned in the paper. The paper does not focus on the final application; instead, it introduces an uncertainty quantification method and demonstrates its potential.
>
> Q2: The decomposition of aleatoric and epistemic uncertainties is theoretically grounded and agnostic to specific modeling choices. The paper highlights this by utilizing different ensembles.

---

> ### Comment · Reviewer_5JQ7 · 2024-12-02
>
> Thank you for your response. I have decided to maintain my score.

---

### Meta-Review · Area_Chair_ZML3 · 2024-12-19

**Metareview:**

This paper studies the problem of uncertainty quantification in the task of trajectory prediction. The approach centers around a decomposition of the epistemic uncertainty into total and aleatoric components. Experiments are conducted to measure the relationship between uncertainty and prediction error in a variety of autonomous driving scenes, as well as study the impact of modeling choices on the calibration of uncertainty.

### Strengths
Reviewers mentioned that the paper is clear and comprehensively covers related work, that the results are interesting, that the method achieves a more precise estimate of epistemic uncertainty than related work,

### Weaknesses
Reviewers mentioned that the paper doesn't provide strategies for minimizing prediction error, a need to justify why identifying OOD scenarios is useful relative to targeting high-error scenarios, the analysis is limited to a small set of trajectory prediction models, the computational burden, similarities to prior work that decompose prediction uncertainties, lacking details of comparisons, the section on detecting OOD scenarios lacks a clear takeaway, **the body of the paper appears to be substantially similar to another paper currently under review for IEEE ICRA 2025**, and could be strengthened with expanding comparison to other datasets, and lacks practical integration with planning systems.

Overall the reviews point out a variety of weaknesses and the reviewers are not very positive about the paper. Taken together with the fact that Reviewer 7w8N mentioned that the paper is substantially similar to a paper under review for IEEE ICRA 2025 (the authors didn't respond to this claim), I don't think there's a clear case for acceptance. The paper needs to expand its evaluation, clarify its contributions, and expand its comparisons to related work.

**Additional Comments On Reviewer Discussion:**

Limited discussion focused on the weaknesses of the paper. Only one of the reviewers meaningfully engaged with the author responses, although almost all reviewers responded. The reviewer that meaningfully engaged with the author responses revised their rating from 3 to 5, and remarked that the paper should include an analysis of the effect of the number of samples used and a more in-depth analysis of its limitations -- across driving datasets and scenarios and an analysis of sensitivities.

---

### Decision · Program_Chairs · 2025-01-22

Reject